**Brief Communication**

# Efficiently accelerated bioimage analysis with NanoPyx, a Liquid Engine-powered Python framework

Bruno M. Saraiva[1,2,12], Inês Cunha[1,3,4,12], António D. Brito [1,5,12], Gautier Follain [6], Raquel Portela[5], Robert Haase [7], Pedro M. Pereira [5], Guillaume Jacquemet [6,8,9,10] & Ricardo Henriques [1,2,5,11]✉

The expanding scale and complexity of microscopy image datasets require accelerated analytical workflows. NanoPyx meets this need through an adaptive framework enhanced for high-speed analysis. At the core of NanoPyx, the Liquid Engine dynamically generates optimized central processing unit and graphics processing unit code variations, learning and predicting the fastest based on input data and hardware. This data-driven optimization achieves considerably faster processing, becoming broadly relevant to reactive microscopy and computing fields requiring efficiency.

Super-resolution microscopy has revolutionized cell biology by enabling fluorescence imaging at an unprecedented resolution[1–4]. However, data collected from these experiments often require specific analytical procedures, such as image registration, resolution enhancement and quantification of data quality and resolution. Many of these procedures use open-source image analysis software, particularly ImageJ[5]/FIJI[6] or napari[7]. The computational performance of each of these tools bears notable implications for processing time, which becomes especially salient given the increasing need for high-performance computing in bioimaging analysis. In this work we present NanoPyx, a Python framework for microscopy image analysis that exploits the Liquid Engine to massively accelerate analysis workflows.

With the increasing use of deep learning, many bioimaging analysis pipelines are now being developed in Python. Pure Python code often runs on a single central processing unit (CPU) core, impacting the performance and speed of Python frameworks. Alternative solutions, such as Cython[8], PyOpenCL[9] and Numba[10], allow CPU and graphics processing unit (GPU) parallelization, which can reduce run times (Supplementary Note 1). However, identifying the swiftest implementation depends on the hardware, input data and parameters.

Figure 1 illustrates a case where denoising the larger image with a nonlocal means (NLM) algorithm[11,12] is approximately two times faster when using a CPU unthreaded strategy than a pixel-wise threaded implementation strategy on a GPU in a professional workstation (Fig. 1c and Supplementary Note 2). Notably, the same algorithm cannot be run on the testing laptop's GPU with the same parameters due to architecture limitations (Fig. 1b). This means that certain acceleration strategies have hardware constraints and require a different approach. However, for other conditions (condition 2 on workstation and laptop and condition 3 on laptop), GPU-based processing is a faster alternative for the same NLM algorithm. Extended Data Figs. 1–5 further support these observations, by illustrating run times for various implementations across distinct datasets and parameters on contrasting hardware set-ups. Another example is Catmull–Rom[13] interpolations parallelized in a pixel-wise manner (Extended Data Fig. 2), in which choosing an OpenCL[14] implementation for lower-sized images could escalate run time by several orders of magnitude compared with parallelized CPU processing. Similarly, threaded CPU processing for larger-sized images performed up to 30 times more slowly than GPU processing on professional workstations. Supplementary Tables 1–4 present benchmarks

[1]Instituto Gulbenkian de Ciência, Oeiras, Portugal. [2]Gulbenkian Institute for Molecular Medicine, Oeiras, Portugal. [3]Instituto Superior Técnico, Lisbon, Portugal. [4]Science for Life Laboratory, Department of Biochemistry and Biophysics, Stockholm University, Stockholm, Sweden. [5]Instituto de Tecnologia Química e Biológica António Xavier, Universidade Nova de Lisboa, Oeiras, Portugal. [6]Turku Bioscience Centre, University of Turku and Åbo Akademi University, Turku, Finland. [7]DFG Cluster of Excellence "Physics of Life", TU Dresden, Dresden, Germany. [8]Turku Bioimaging, University of Turku and Åbo Akademi University, Turku, Finland. [9]Faculty of Science and Engineering, Cell Biology, Åbo Akademi University, Turku, Finland. [10]InFLAMES Research Flagship Center, Åbo Akademi University, Turku, Finland. [11]UCL-Laboratory for Molecular Cell Biology, University College London, London, UK. [12]These authors contributed equally: Bruno M. Saraiva, Inês Cunha, António D. Brito. ✉e-mail: r.henriques@itqb.unl.pt

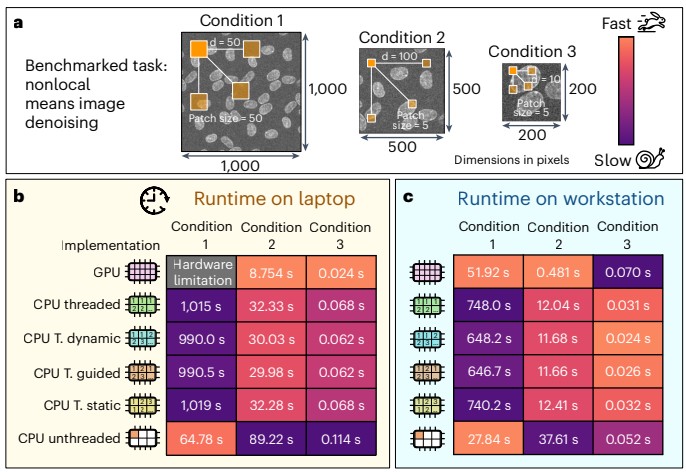

**Fig. 1 | Comparative run times of multiple implementations of an algorithm, run on a consumer-grade laptop and a professional workstation. a–c,** The fastest implementation (Supplementary Note 3) depends on various factors such as the shape of the input data, method-specific parameters and the user device. **a,** Nonlocal mean denoising is performed on images of varying shape using a collection of patch sizes and distances (d). **b,** Run times of various conditions when performing analysis on a consumer-grade laptop; condition 1 could not be run on the GPU due to hardware limitations. T. dynamic, threaded dynamic; T. guided, threaded guided; T. static, threaded static. **c,** On a professional workstation, faster implementation changes with each condition, illustrating how it is affected by the input data and method-specific parameters.

across ten different hardware set-ups, highlighting the limitations of relying on a single implementation, because it may not universally offer the fastest performance.

Here we introduce NanoPyx, a high-performance bioimaging analysis framework exploiting the Liquid Engine. It uses multiple variations (here called implementations; Supplementary Note 3) of the same algorithm to perform a specific task. These variations include multiple acceleration strategies, including PyOpenCL[9], CUDA[15] (using CuPy[16]), Cython[8], Numba[10], Transonic[17] and Dask[18] (Extended Data Figs. 1–5). Although these implementations provide numerically identical outputs for the same input, their computational performance differs by exploiting different computational strategies. The Liquid Engine features three main components: (1) metaprogramming tools for multihardware implementation (using Mako templates[19] and a custom script, named c2cl; Supplementary Note 4); (2) an automatic benchmarking system; and (3) a supervisor machine learning-based agent that determines the ideal combination of implementations to maximize performance (Fig. 2).

Liquid Engine uses a machine learning system (Supplementary Note 5) to predict the optimal combination of implementations while including device-dependent performance variations (Fig. 1 and Extended Data Figs. 1–5). When a user does not have access to one of the implementations, Liquid Engine ignores it, guaranteeing that the user will always be able to process their images. Dynamic benchmarking substantially enhances computational speed for tasks involving input data of varying size. This technique predicts when to switch between different algorithmic implementations, resulting in up to 24-fold faster processing compared with use of the pixel-wise parallelization strategy (CPU threaded; Fig. 2). Even when compared with running both methods on a GPU, performing dynamic implementation selection still provides 1.75-fold acceleration (Supplementary Table 5).

Liquid Engine maintains a historic record of run times for each implementation. Manual benchmarking can be initiated by the user, prompting Liquid Engine to profile the execution of each implementation and identify the fastest (Supplementary Table 5). The system uses fuzzy logic[20] (Supplementary Note 6) to identify the benchmarked example with the most similar input properties, utilizing it as a baseline for the expected

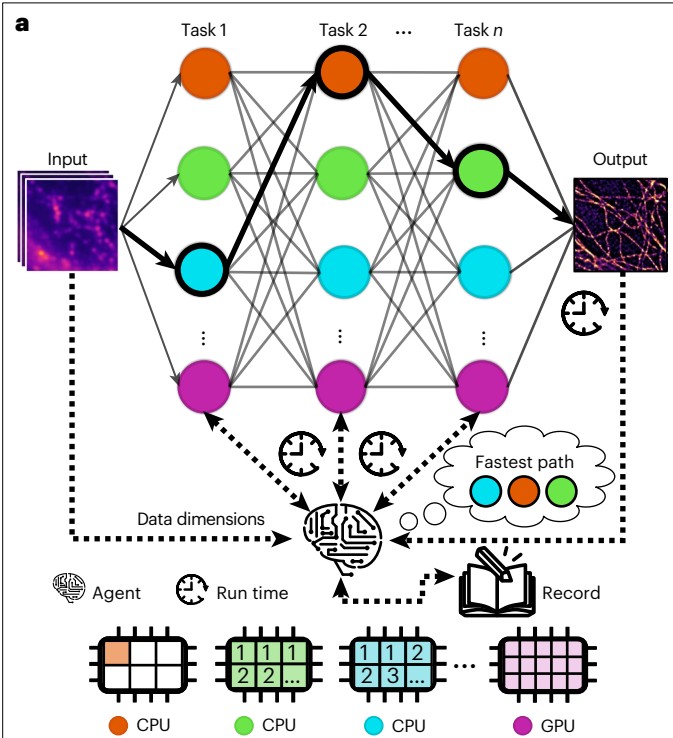

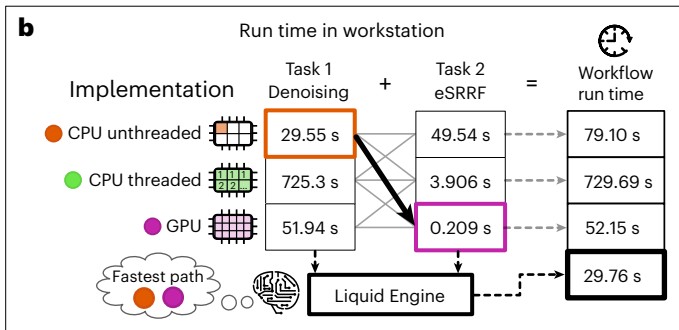

**Fig. 2 | NanoPyx achieves optimal performance by exploiting Liquid Engine's self-optimization capabilities. a,** NanoPyx is built on top of the Liquid Engine, which automatically benchmarks implementations of all tasks in a specific workflow. Liquid Engine retains a historical record of the run times of each task and input used, allowing a machine learning-based agent to select the fastest combination of implementations. **b,** Liquid Engine dynamically chooses the fastest implementation for each method, based on its input parameters. For a workflow performing denoising on a 1,000 × 1,000 image, using NLM[11,12] (patch distance 50 pixels, patch size 50 pixels, sigma 1.0 and cut-off distance (h) 0.1), followed by super-resolution of the data with eSRRF[21] (magnification ×5, radius 1.5, sensitivity 1 and using intensity weighting), Liquid Engine selects the fastest combination of implementations to substantially reduce run times.

execution time (Supplementary Table 5). This system enables NanoPyx to make instant decisions based on an initially limited set of records, progressively improving its performance as further data are obtained.

Each time a workflow is scheduled to run, a supervisor agent is responsible for selecting the best implementation based on previous run times; this selection is made without imposing any substantial overhead (Supplementary Table 5). When users do not trigger manual benchmarking, the agent uses 'factory-default' benchmarks until sufficient run times have been recorded on the user's hardware. The agent constantly monitors the run times of all available methods, and can adapt to unexpected delays by ensuring that the optimal implementation is selected. In the case where a severe delay is detected, the agent predicts whether the optimal implementation has changed and calculates the likelihood of that delay being repeated in the future (Extended Data Fig. 6). Over the course of

several sequential runs of the same method, we show that delay management improved average run time by a factor of 1.8 for a two-dimensional (2D) convolution and by 1.5 for an established super-resolution radial fluctuations (eSRRF)[21] analysis (Extended Data Fig. 7).

NanoPyx enhances and expands the super-resolution analysis methods previously included in the NanoJ[21–24] plugin family, and introduces additional bioimage analysis techniques, including example testing datasets (Supplementary Note 7). Extended Data Fig. 8 illustrates an example workflow where NanoPyx starts by performing drift correction. NanoPyx then allows super-resolution reconstructions using SRRF[23] or its improved version, eSRRF[21]. Next, quality assessment is performed by running Fourier ring correlation[25] and decorrelation analysis[26], and by calculating a SQUIRREL error map[24]. Besides the aforementioned methods, NanoPyx also includes channel registration (Extended Data Fig. 9), multiple interpolators, 2D convolution, denoising through NLM[11,12] and several other bioimage analysis methods (Supplementary Table 6). Although not all of these methods exploit the advantages of Liquid Engine (Supplementary Table 6), we are actively developing new parallelization strategies for the remaining methods.

NanoPyx is accessible as a Python library, which can be installed via either Python package index or our GitHub repository (Supplementary Table 5). Liquid Engine is also available as a standalone Python package that is readily integrated in other projects. Alongside these Python libraries, we provide cookiecutter (https://cookiecutter.readthedocs.io) template files to help developers implement their own methods using Liquid Engine (Supplementary Note 8). Secondly, we provide Jupyter notebooks[27] (Supplementary Fig. 1a and Supplementary Table 5). Users of these notebooks are not required to interact with any code directly, because a graphical user interface is generated[28]. Lastly, we developed a plugin for napari[7], a Python image viewer (Supplementary Fig. 1b). By offering these three distinct user interfaces, we ensure that NanoPyx can be readily utilized by users irrespective of their coding proficiency level. In NanoPyx's repository, we have provided usage guidelines for end-users along with several tutorials, including videos (Supplementary Table 7), on how to run NanoPyx through any of its interfaces, and how to implement their own methods exploiting optimization of Liquid Engine (Supplementary Note 8).

Looking ahead, a priority for NanoPyx is expanding support for emerging techniques such as artificial intelligence-assisted imaging and smart microscopes. Because these methods involve processing data in real time during acquisition, NanoPyx's accelerated performance becomes critical. In addition, we aim to incorporate more diverse processing workflows beyond currently implemented methods.

Cumulatively, NanoPyx delivers adaptive performance optimization to accelerate bioimage analysis while retaining modular design and easy adoption. This flexible framework is important and timely, given the expanding volumes of microscopy data and the need for data-driven reactive microscopy. The optimization principles embodied in its Liquid Engine can be extended to other scientific workloads requiring high computational efficiency. As data scales expand, NanoPyx offers researchers an actively improving platform to execute demanding microscopy workflows.

## Online content

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

## Methods

### Mammalian cell culture

Human umbilical vein endothelial cells (HUVEC) (PromoCell, catalog no. C-12203) were grown in endothelial cell growth medium (PromoCell, catalog no. C-22010), with a supplementary mix ((Promocell, catalog no. C-39215) and 1% penicillin/streptomycin (Sigma); Fig. 1). Endothelial primary cells from P0 (commercial vial) were expanded to a P3 stock frozen at −80 °C to standardize the experimental replicates. A549 cells (The European Collection of Authenticated Cell Cultures) were cultured in phenol red-free, high-glucose, L-glutamine containing DMEM (Thermo Fisher Scientific), supplemented with 10% (v/v) fetal bovine serum (Sigma) and 1% (v/v) penicillin/streptomycin (Thermo Fisher Scientific), at 37 °C in an incubator with 5% $CO_2$ (Extended Data Fig. 8).

### Sample preparation for microscopy

HUVEC were seeded in an eight-well, glass-bottom μ-slide (Ibidi, catalog no. 80807) precoated with warm endothelial cell growth medium without antibiotics (50,000 cells per well). Cells were then grown for 48 h, fixed with prewarmed 4% paraformaldehyde in PBS (Thermo Fisher Scientific, catalog no. 28908) for 10 min at 37 °C and stained with DAPI. A549 cells were seeded on an eight-well, glass-bottom μ-slide (ibidi) at density $0.05–0.10 × 10^6$ cells $cm^{-2}$. Following 24 h incubation at 37 °C and under 5% $CO_2$, cells were washed once with PBS and fixed for 20 min at 23 °C in 4% paraformaldehyde in PBS. Following fixation, cells were washed three times in PBS (5 min each), quenched for 10 min in a solution of 300 mM glycine (in PBS) and permeabilized using a solution of 0.2% Triton-X (in PBS) for 20 min at 23 °C. Following three washes (5 min each) in washing buffer (0.05% Tween-20 in PBS), cells were blocked for 30 min in blocking buffer (5% BSA and 0.05% Tween-20 in PBS). Samples were then incubated with a mix of anti-α-tubulin antibodies (1 μg $ml^{-1}$ clone DM1A (Sigma), 2 μg $ml^{-1}$ clone 10D8 (BioLegend), 2 μg $ml^{-1}$ clone AA10, BioLegend) and anti-septin 7 (1 μg $ml^{-1}$, catalog no. 18991, IBL) for 16 h at 4 °C in blocking buffer. Following three washes (5 min each) in washing buffer, cells were incubated with Alexa Fluor 647 conjugated goat anti-mouse IgG and Alexa Fluor 555 conjugated goat anti-rabbit IgG (6 μg $ml^{-1}$ in blocking buffer) for 1 h at 23 °C. Cell nuclei were counterstained with Hoechst 33342 (1 μg $ml^{-1}$). Cells were then washed three times (5 min each) in washing buffer and once in 1× PBS for 10 min. Finally, cells were mounted using glucose oxidase and β-mercaptoethylamine (50 mM Tris, 10 mM NaCl, pH 8.0, supplemented with 50 mM β-mercaptoethylamine, 10% (w/v) glucose, 0.5 mg $ml^{-1}$ glucose oxidase and 40 μg $ml^{-1}$ catalase).

### Data acquisition

HUVEC were imaged using a Marianas spinning-disk confocal microscope equipped with a Yokogawa CSU-W1 scanning unit on an inverted Zeiss Axio Observer Z1 microscope, controlled by SlideBook 6 (Intelligent Imaging Innovations, Inc.) (Fig. 1). Images were acquired using an Evolve 512 EMC CD camera (chip size, 512 × 512; Photometrics); the objective used was an M27 ×63/1.4 numerical aperture (NA), oil immersion (Plan-Apochromat). Data acquisition was performed with a Nanoimager microscope (Oxford Nanoimaging) equipped with an Olympus ×100/1.45 NA oil-immersion objective (Extended Data Fig. 8). Imaging was performed using 405-, 488- and 640-nm lasers for Hoechst-33342, AlexaFluor555 and AlexaFluor647 excitation, respectively. Fluorescence was detected using a sCMOS camera (ORCA Flash, 16 bit). For channel 0, a dichroic filter with bands of 498–551 and 576–620 nm was used and, for channel 1, a 665–705-nm dichroic filter. Sequential multicolor acquisition was performed for AlexaFluor647, AlexaFluor555 and Hoechst-33342. Using epifluorescence illumination, a pulse of high laser power (90%) of the 640-nm laser was used, with 10,000 frames immediately acquired. The sample was then excited with the 488-nm laser (13.7% laser power), with 500 frames acquired, followed by 405-nm laser excitation (40% laser power) with acquisition of a further 500 frames. For all acquisitions, an exposure time of 10 ms was used.

### Liquid Engine agent

Run times of methods implemented in NanoPyx through Liquid Engine are locally stored on the user's home folder inside a folder titled .liquid_engine. For OpenCL implementations, the agent also stores an identification of the device and can detect hardware changes. Whenever a method is run through Liquid Engine, the overseeing agent reads the 50 most recent recorded run times. If there are fewer than 50 recorded runs but more than three, the agent will proceed with the available recorded runs. However, if there are fewer than three runs recorded, all Liquid Engine methods will revert to default benchmarks that can be either supplied with the package or defined by the user. For each implementation, the agent then divides the available corresponding run times into two separate sets of equal length, one containing the fastest run times and the other the slowest. We then calculate average and standard deviation for both sets, namely FastAverage, FastStdDev, SlowAverage and SlowStdDev (equations (1–5)). This split in run times helps identify the start or end of a delay. By comparison against the set of fastest run times, we ensure that previous delayed run times do not skew normal average run time. On the other hand, the set of slowest run times, although not guaranteed to be exactly like a delayed run time, helps us estimate a lower bound to that which a higher-than-average run time could look like.

Once the method has finished running, the agent checks whether there was a delay (Delay). A delayed implementation is defined by having its run time (Measured Run Time) higher than the recorded average run time of the fastest runs, plus four times the standard deviation of the fastest runs (equation (1)). If a delay is detected (Extended Data Fig. 6), the agent will also calculate the delay factor (DelayFactor, equation (2)) and will activate a probabilistic approach that stochastically selects which method to run.

This is performed using a logistic regression model that calculates the probability of the delay being present on the next run ($P_{delay}$), and by adjusting the expected run time of the delayed implementation (Adjusted Run Time) according to equation (3), while still using FastAverage for all nondelayed implementations. The agent then picks which implementation to use, based on probabilities assigned to each implementation (given by $P_{Run Time\,k}$ for a given implementation $k$), using 1 over the square of adjusted run time and normalized for the run times of all other implementations (equation (4)). This stochastic approach ensures that the agent will still run the delayed implementation from time to time to check whether that delay is still present.

During a subsequent run, the agent will evaluate whether there is a delay. It will consider the delay as over when the measured run time is either (1) lower than the slow average minus the standard deviation (Std) of the slowest runs, or (2) lower than the fast average plus the standard deviation of the fastest runs (as per equation (5)). Once the delay is over, the agent will revert to selecting which implementation to use based on the fast average of each implementation (as shown in Extended Data Figs. 6 and 7).

$$\text{Delay} = \text{True if Measured Run Time} > (\text{FastAverage} + 4 \times \text{Std}) \quad (1)$$

$$\text{DelayFactor} = \frac{\text{Measured Run Time}}{\text{FastAverage}} \quad (2)$$

Adjusted Run Time

$$= \text{FastAverage} \times (1 - P_{delay}) + \text{FastAverage} \times \text{DelayFactor} \times P_{delay} \quad (3)$$

$$P_{\text{Run Time}\,k} = \sum \frac{1}{\text{Run Time}^2} \times \frac{1}{\text{Adjusted Run Time}_k^{2^2}} \quad (4)$$

$$\text{Delay} = \text{False if}(\text{Measured Run Time} < (\text{SlowAverage} - \text{SlowStdDev}))$$
$$\lor (\text{Measured Run Time} > (\text{FastAverage} + \text{FastStdDev})) \tag{5}$$

## Benchmarking run times

For laptop benchmarks, a MacBook Air M1 Pro with 16 GB of random-access memory (RAM) and a 512-GB, solid-state drive (SSD) was used. For the professional workstation, a custom-made desktop computer was used containing an Intel i9-13900K, a NVIDIA RTX 4090 with 24 GB of dedicated video memory, a 1 TB SSD and 128 GB of DDR5 RAM. The first benchmark performed (Fig. 1 and Extended Data Fig. 2) was a fivefold upsampling of the input data, using a Catmull–Rom[13] interpolator. Benchmarks were performed on three different input images with shapes of $1 \times 10 \times 10$, $10 \times 10 \times 10$, $10 \times 100 \times 100$, $10 \times 300 \times 300$, $100 \times 300 \times 300$ and $500 \times 300 \times 300$ (time points × height × width). The second benchmarks (Extended Data Fig. 1) were nonlocal means denoising performed on images of $200 \times 200$, $500 \times 500$ and $1,000 \times 1,000$ pixels using, respectively, 10, 100 and 50 as patch distance, with varying patch size (5, 10, 20, 50 and 100). The third benchmarks (Extended Data Figs. 2–5) were 2D convolutions using a kernel of varying size (1, 5, 9, 13, 17, 21), where all kernel values are 1, on images of varying size (100, 500, 1,000, 2,500, 5,000, 7,500, 10,000, 15,000 or 20,000 pixels for both dimensions). Supplementary Tables 1–4 describe ten different hardware set-ups used for benchmarking three different conditions of 2D convolution, Catmull–Rom interpolation and nonlocal means denoising.

## Benchmarking delay management

For evaluation of Liquid Engine's delay management capabilities, we benchmarked its performance on 2D convolutions and eSRRF reconstructions under induced delay conditions. The hardware used was a high-end desktop with an Intel i9-13900K CPU, NVIDIA RTX 4090 GPU, 128-GB DDR5 RAM and 1-TB SSD. For the 2D convolution task, we applied a $9 \times 9$ kernel on $6,000 \times 6,000$-pixel random images. To simulate a delay, we used a separate Python process that allocated >24 GB of GPU memory for irrelevant computations, thus overloading the GPU. We executed 400 sequential convolutions, introducing artificial delay during convolutions 101–200, and compared run times with and without Liquid Engine optimization enabled. Similarly, for eSRRF, we reconstructed a $100 \times 100 \times 100$-pixel random volume with parameters magnification = 5, radius = 1.5 and sensitivity = 1. Artificial delay was induced on reconstructions 51–100 out of a total of 200. Run times were again collected and analyzed with Liquid Engine on and off. In both tasks, Liquid Engine detected abnormal delay during the overloaded period based on run time spikes; it then switched its implementation preference probabilistically to avoid using the delayed GPU code.

## Reporting summary

Further information on research design is available in the Nature Portfolio Reporting Summary linked to this article.

## Data availability

The datasets used in the figures are either listed in Supplementary Table 8 or are available for download via Zenodo at https://zenodo.org/record/8318395 (ref. 29). Source data are provided with this paper.

## Code availability

The NanoPyx python library and Jupyter Notebooks can be found in our Github repository (https://github.com/HenriquesLab/NanoPyx). The Liquid Engine Python library can be found in our GitHub repository (https://github.com/HenriquesLab/LiquidEngine). The cookiecutter templates can be found in this GitHub repository (https://github.com/HenriquesLab/LiquidEngineCookieCutter). The napari plugin implementing all NanoPyx methods can be found in a separate Github repository (https://github.com/HenriquesLab/napari-NanoPyx).

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

## Acknowledgements

We thank the previous developers of the NanoJ framework, whose work inspired this study. In addition, we thank L. Royer and J. Nunez-Iglesias for their invaluable feedback and guidance in preparing our work. R. Henriques, P.M.P. and R.P. acknowledge support from LS4FUTURE Associated Laboratory (no. LA/P/0087/2020). R. Henriques, B.M.S. and I.C. acknowledge the support of the Gulbenkian Foundation (Fundação Calouste Gulbenkian); the European Research Council under the European Union's Horizon 2020 research and innovation program (grant agreement no. 101001332); the European Union through the Horizon Europe program (AI4LIFE project with grant agreement no. 101057970-AI4LIFE and RT-SuperES project with grant agreement no. 101099654-RT-SuperES); the European Molecular Biology Organization Installation Grant (no. EMBO-2020-IG-4734); and the Chan Zuckerberg Initiative Visual Proteomics Grant (no. vpi-0000000044; https://doi.org/10.37921/743590vtudfp). In addition, A.D.B. acknowledges the FCT 2021.06849.BD fellowship. R. Henriques and B.M.S. also acknowledge that this project has been made possible in part by a grant from the Chan Zuckerberg Initiative DAF, an advised fund of Silicon Valley Community Foundations (Chan Zuckerberg Initiative Napari Plugin Foundations Grants Cycle 2, no. NP2-0000000085). P.M.P. and R.P. acknowledge support from Fundação para a Ciência e Tecnologia (Portugal) project grant no. PTDC/BIA-MIC/2422/2020 and the MOSTMICRO-ITQB R&D Unit (nos. UIDB/04612/2020 and UIDP/04612/2020). P.M.P. acknowledges support from La Caixa Junior Leader Fellowship (no. LCF/BQ/PI20/11760012), financed by 'la Caixa' Foundation (ID100010434) and the European Union's Horizon 2020 research and innovation program under Marie Skłodowska-Curie grant agreement no. 847648, and from a Maratona da Saúde award. This study was supported by the Academy of Finland (no. 338537 to G.J.), the Sigrid Juselius Foundation (to G.J.), the Cancer Society of Finland (Syöpäjärjestöt, to G.J.) and the Solutions for Health strategic funding to Åbo Akademi University (to G.J.). This research was supported by InFLAMES Flagship Program of the Academy of Finland (decision no. 337531).

## Author contributions

B.M.S., P.M.P., G.J. and R. Henriques conceived the study in its initial form. B.M.S., I.C., A.D.B. and R. Henriques developed the NanoPyx framework, with code contributions from R. Haase and G.J. B.M.S., I.C., A.D.B. and R. Henriques designed the Liquid Engine optimization approach. B.M.S., I.C. and A.D.B. implemented the Liquid Engine tools. G.F., R.P., P.M.P. and G.J. provided samples, data, critical feedback, testing and guidance. B.M.S., I.C., A.D.B., G.F. and G.J. performed experiments and analysis. B.M.S., P.M.P., G.J. and R. Henriques acquired funding. B.M.S., P.M.P., R. Haase, G.J. and R. Henriques supervised the work. B.M.S., I.C., A.D.B., G.J. and R. Henriques wrote the manuscript, with input from all authors.

## Competing interests

The authors declare no competing interests.

## Additional information

**Extended data** is available for this paper at https://doi.org/10.1038/s41592-024-02562-6.

**Correspondence and requests for materials** should be addressed to Ricardo Henriques.

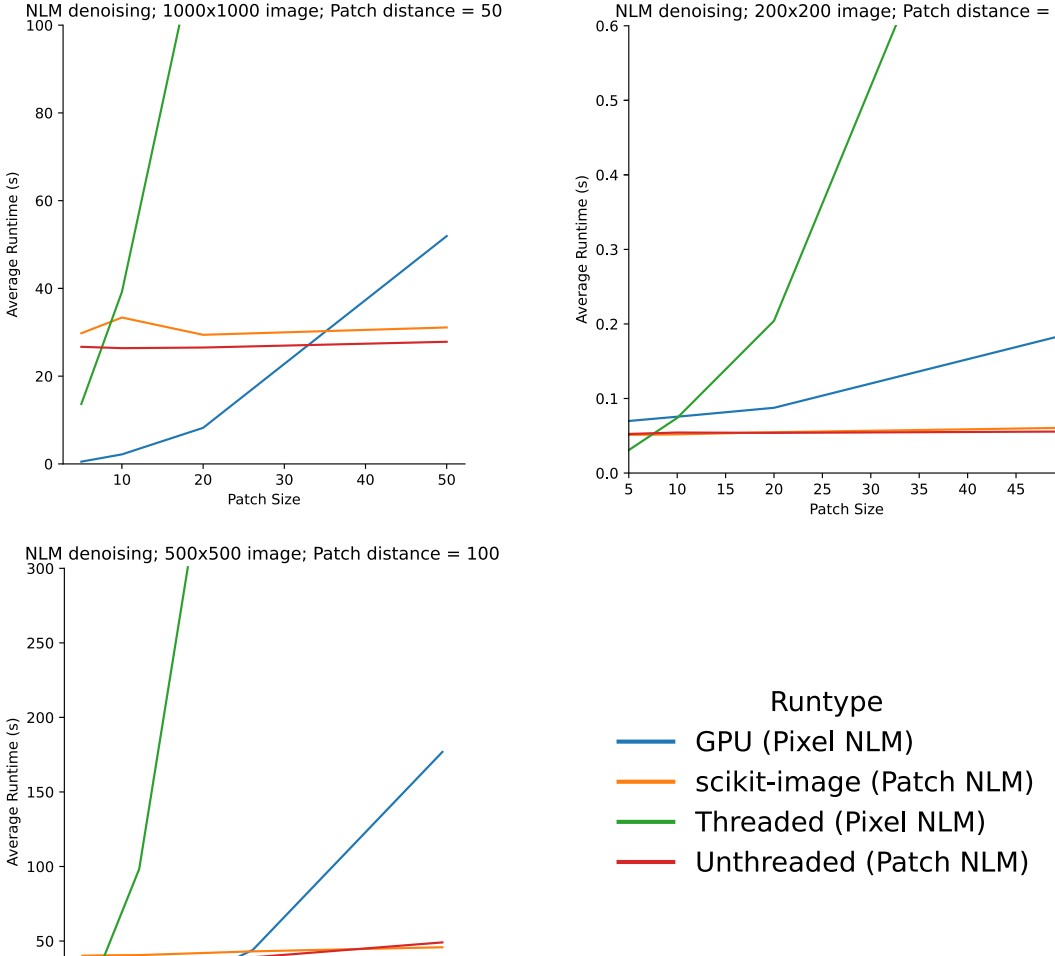

**Extended Data Fig. 1 | Run times of non-local means denoising are dependent on their implementation.** The fastest run time for non-local means denoising changes according to the parameters defined. The pixel-wise implementation thrives when both the patch distance and patch size is relatively small.

The patch-wise implementation is virtually independent on the patch size and although it sports higher memory costs its computational efficiency makes it an attractive option for bigger patch sizes.

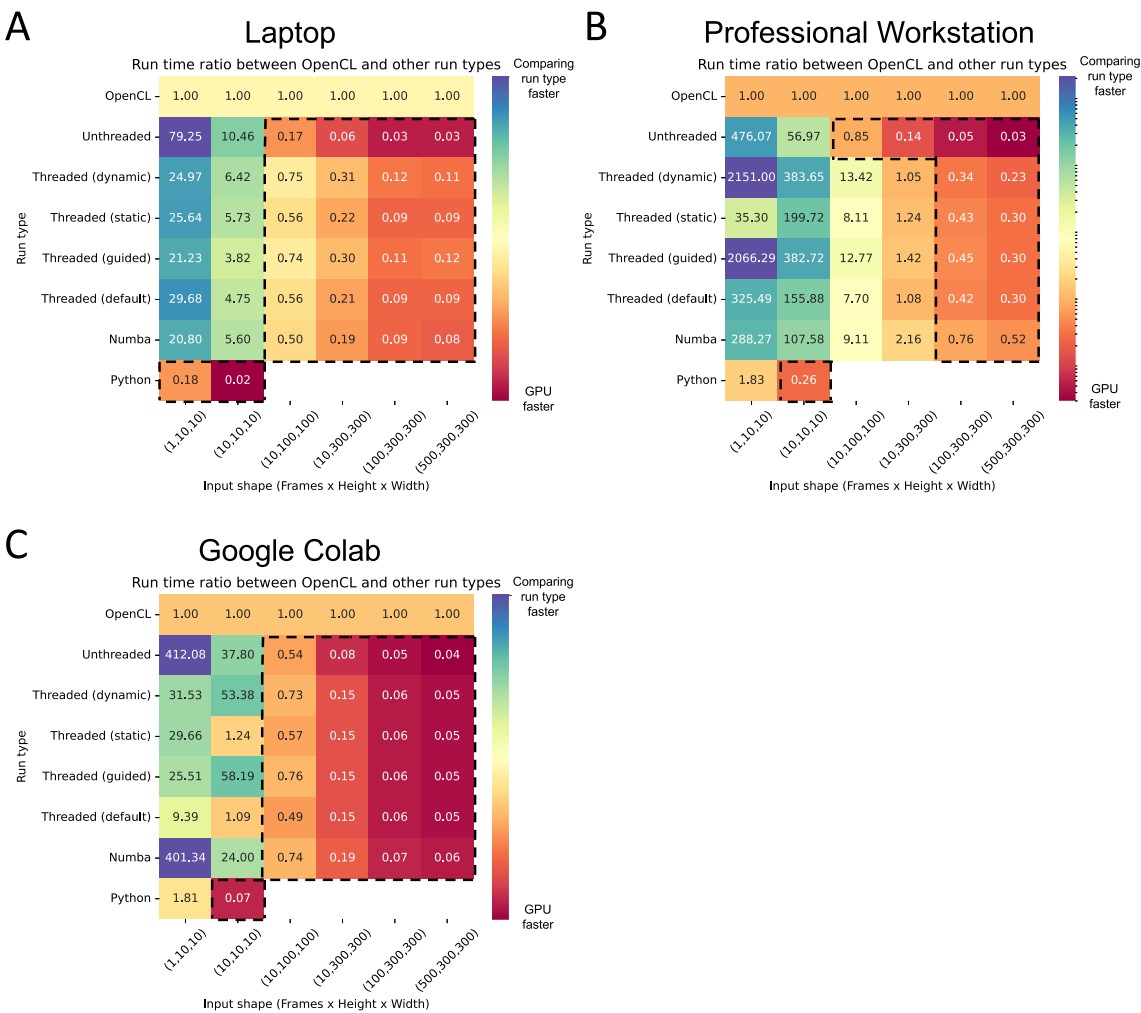

**Extended Data Fig. 2 | Ratio between the run times of OpenCL and other implemented run types.** Run times of a 5x Catmull-rom[13] interpolation were measured across multiple input data sizes using either a MacBook Air M1 (**A**), a Professional Workstation (**B**) or Google Colaboratory (**C**). Using the fastest implementation can lead to up to 10x faster code execution. Area within dashed lines correspond to kernel and image sizes where OpenCL is faster than other implementations.

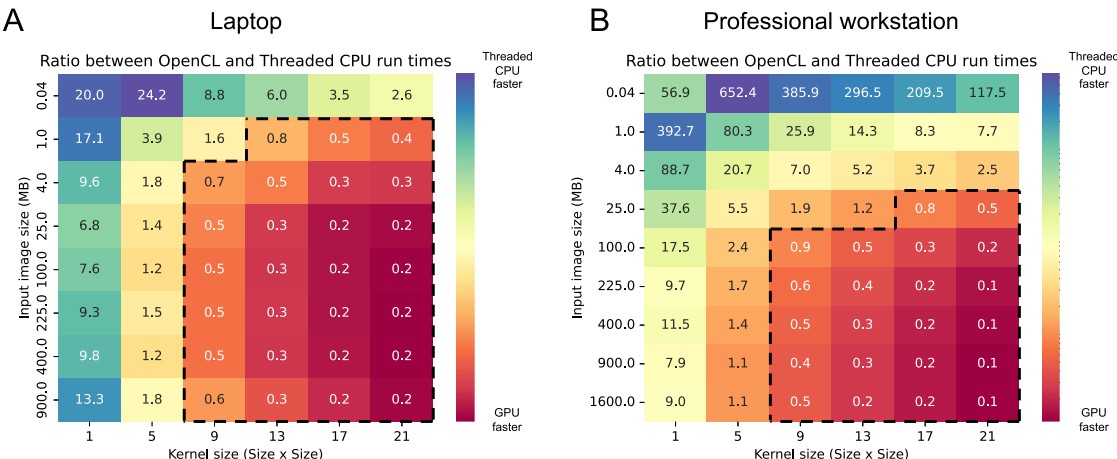

**Extended Data Fig. 3 | Ratio between the run times of a 2D convolution.** Run times were measured across multiple input data sizes and kernel sizes using either a MacBook Air M1 (**A**) or a Professional Workstation (**B**). Areas within dashed lines correspond to kernel and image sizes where OpenCL is faster than threaded CPU.

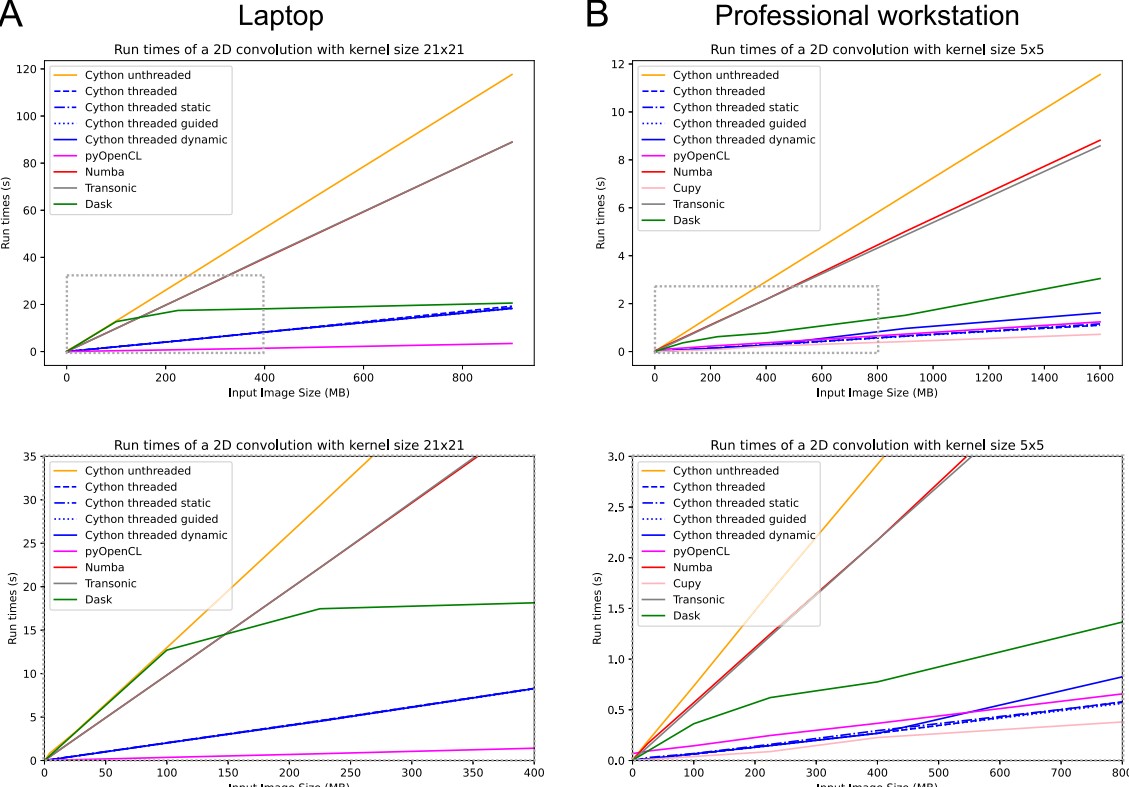

**Extended Data Fig. 4 | Run time of each implementation is dependent on the shape of input data.** A 2D convolution was performed on images with increasing size using either a MacBook Air M1 Pro (**A**) or a professional workstation (**B**). A 21 by 21 kernel was used for the laptop and a 5x5 for the workstation. The run times of each implementation vary according to the size of the input image. Bottom panels correspond to zoomed in windows of top panels, indicated by dotted boxes.

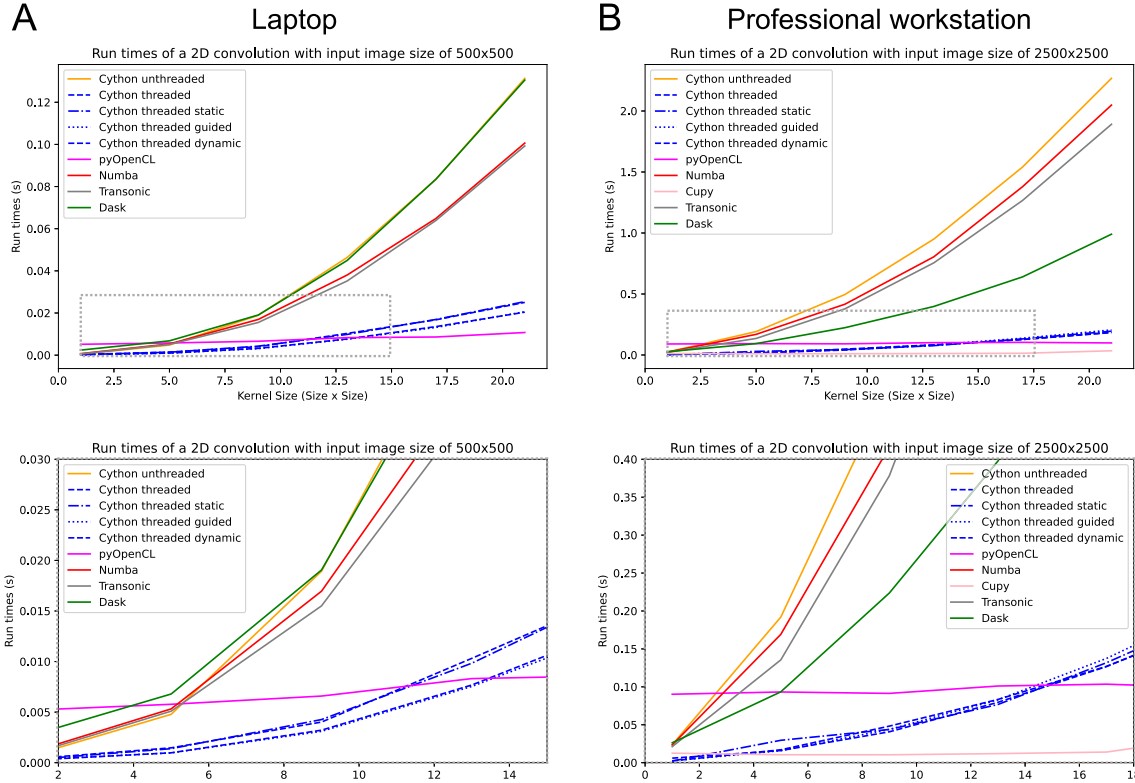

**Extended Data Fig. 5 | Kernel size impacts which implementation is the fastest.** A 2D convolution was performed on images with varying kernel sizes, ranging from 1 to 21 (every 4) using either a MacBook Air M1 Pro on a 500x500 image (**A**) or a professional workstation on 2500x2500 image (**B**). While unthreaded is virtually always the slowest implementation, the threaded implementations are only the fastest for small kernel sizes, after which a GPU-based implementation becomes the fastest. Bottom panels correspond to zoomed in windows of top panels, indicated by dotted boxes.

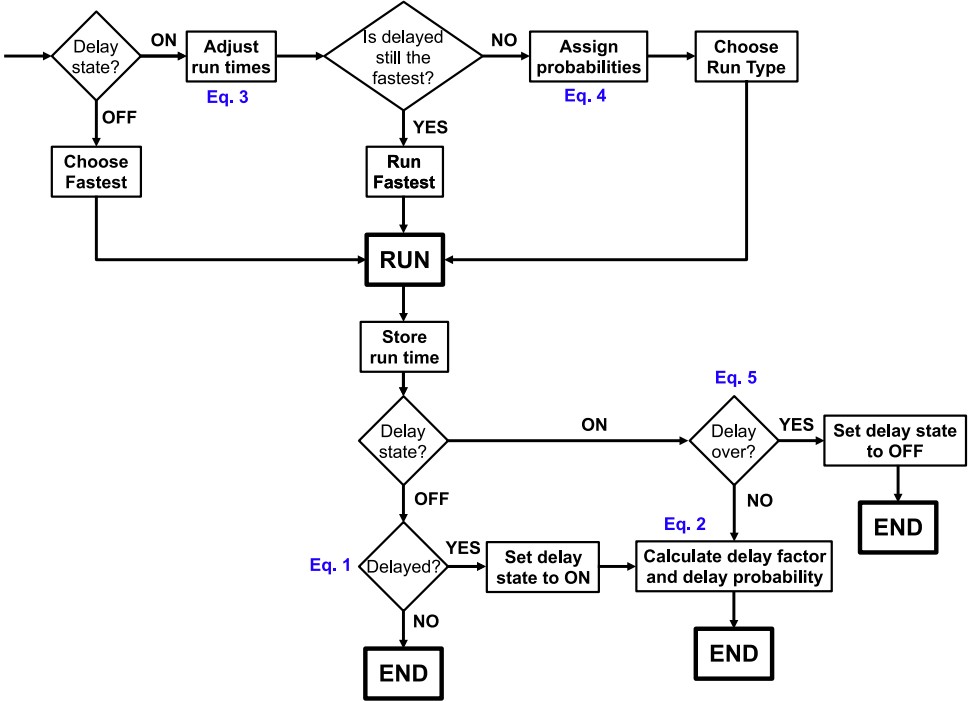

**Extended Data Fig. 6 | Schematic of the Agent decision making for delay management.** The agent identifies delays when an implementation's run time exceeds the fastest average plus four standard deviations (Equation 1). Upon detection, it calculates a delay factor (Equation 2) and uses a probabilistic approach with Logistic Regression to adjust run times (Equation 3) and select implementations stochastically (Equation 4). This ensures delayed implementations are periodically testes while favoring faster alternatives. A delay is considered resolved when the run time falls below thresholds defined in Equation 5, after which the agent reverts to selecting implementations based on their fast average run times.

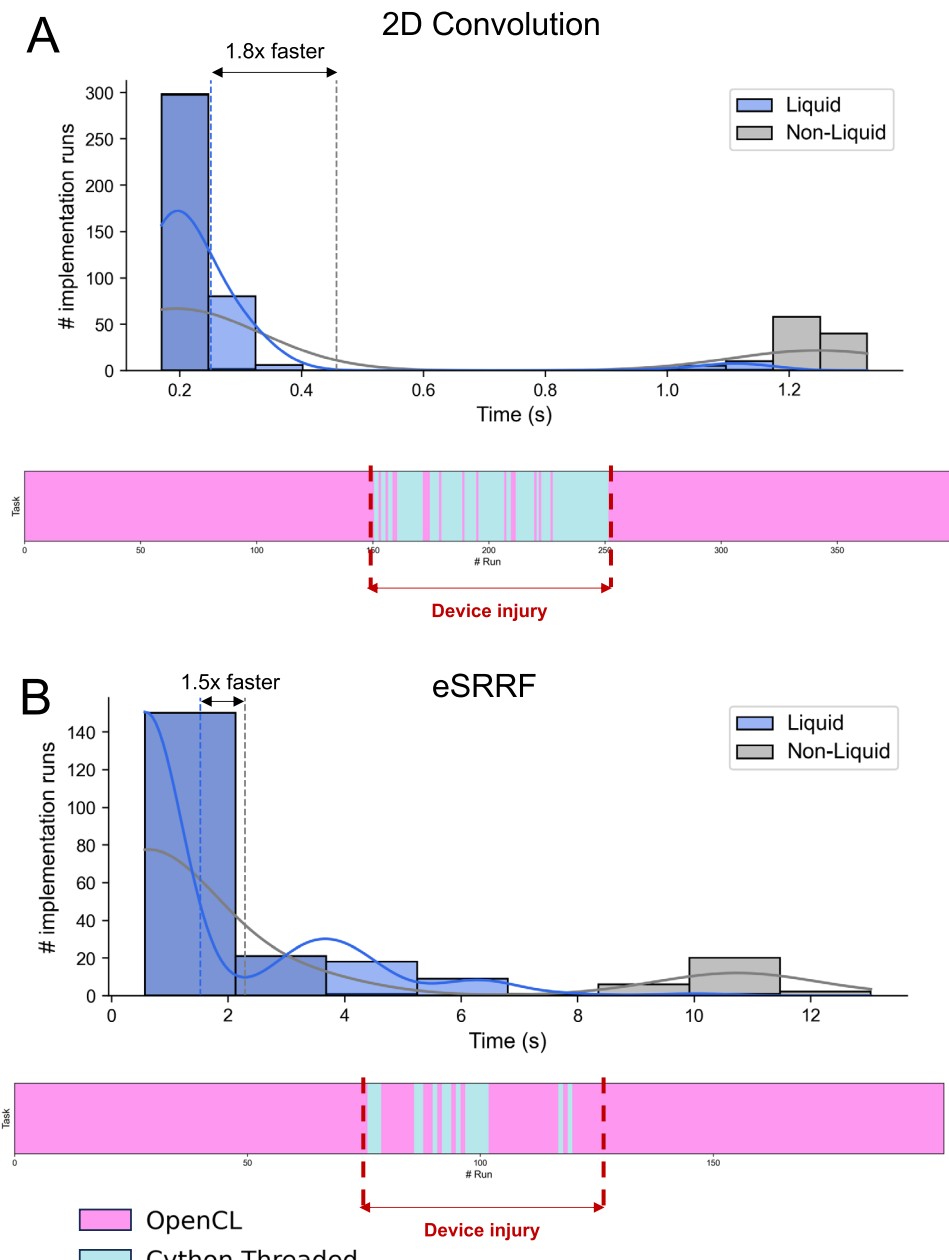

**Extended Data Fig. 7 | Example of delay management by the Liquid engine.**
Multiple two-dimensional convolutions (**A**) and eSRRF analysis (**B**) were run sequentially in a professional workstation. Starting from two initial benchmarks, the Agent is responsible for informing the Liquid Engine on the best probable implementation. An artificial delay was induced by overloading the GPU with superfluous calculations in a separate Python interpreter. Dashed lines represent average run times.

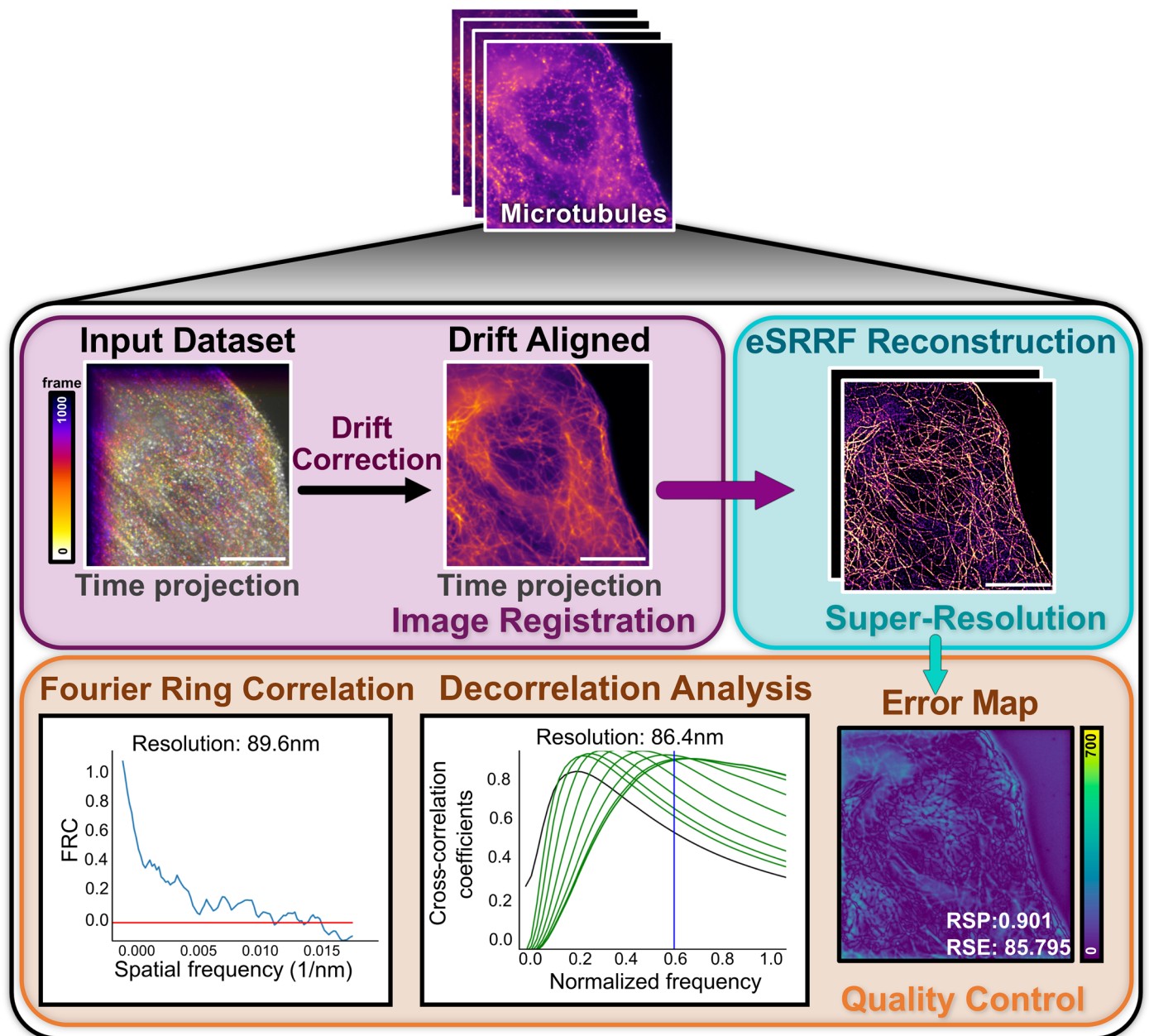

**Extended Data Fig. 8 | Microscopy image processing workflow using NanoPyx methods.** Through NanoPyx, users can correct drift, generate a super-resolved image using eSRRF[21], assess the quality of the generated image using Fourier Ring Correlation (FRC)[25], Image Decorrelation Analysis[26], and NanoJ-SQUIRREL[24] metrics. NanoPyx methods are made available as a Python library, Jupyter Notebooks that can be run locally or through Google Colaboratory, and as a napari plugin. Scale bars, 10 µm.

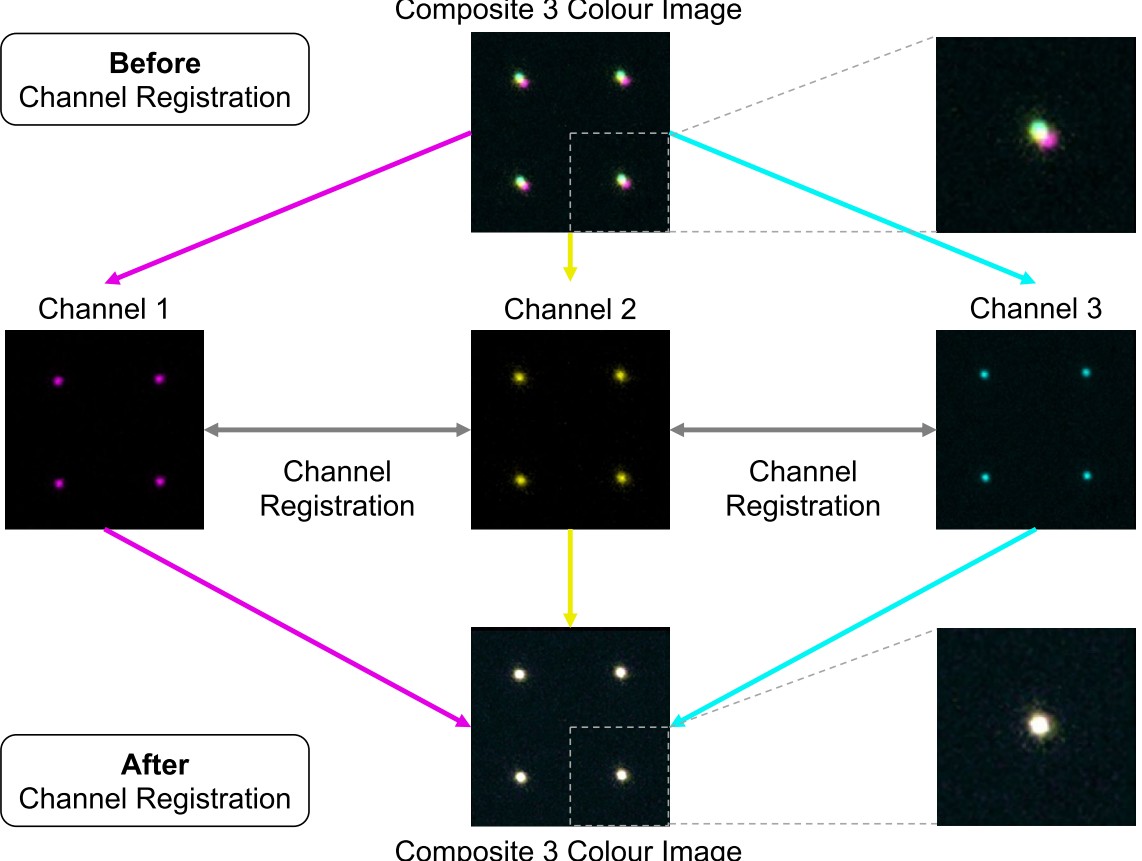

**Extended Data Fig. 9 | Example channel registration of a calibration slide.** NanoPyx allows users to perform channel registration based on the NanoJ[22] implementation. Example data of calibration slide obtained from the freely available data in the Fast4DReg[30] publication.

# Reporting Summary

## Statistics

For all statistical analyses, confirm that the following items are present in the figure legend, table legend, main text, or Methods section.

| n/a | Confirmed | |
|---|---|---|
| ☒ | ☐ | The exact sample size (*n*) for each experimental group/condition, given as a discrete number and unit of measurement |
| ☒ | ☐ | A statement on whether measurements were taken from distinct samples or whether the same sample was measured repeatedly |
| ☒ | ☐ | The statistical test(s) used AND whether they are one- or two-sided<br>*Only common tests should be described solely by name; describe more complex techniques in the Methods section.* |
| ☒ | ☐ | A description of all covariates tested |
| ☒ | ☐ | A description of any assumptions or corrections, such as tests of normality and adjustment for multiple comparisons |
| ☒ | ☐ | A full description of the statistical parameters including central tendency (e.g. means) or other basic estimates (e.g. regression coefficient) AND variation (e.g. standard deviation) or associated estimates of uncertainty (e.g. confidence intervals) |
| ☒ | ☐ | For null hypothesis testing, the test statistic (e.g. *F*, *t*, *r*) with confidence intervals, effect sizes, degrees of freedom and *P* value noted<br>*Give P values as exact values whenever suitable.* |
| ☒ | ☐ | For Bayesian analysis, information on the choice of priors and Markov chain Monte Carlo settings |
| ☒ | ☐ | For hierarchical and complex designs, identification of the appropriate level for tests and full reporting of outcomes |
| ☒ | ☐ | Estimates of effect sizes (e.g. Cohen's *d*, Pearson's *r*), indicating how they were calculated |

*Our web collection on statistics for biologists contains articles on many of the points above.*

## Software and code

Policy information about availability of computer code

| Data collection | Images were collected as indicated in the material and methods. All the instruments used are commercially available and were controlled using the software provided by the manufacturer. |
|---|---|
| Data analysis | Images used in the manuscript were processed with NanoPyx 0.6.1, Napari 0.4.19 or Fiji 1.54f as indicated.<br>NanoPyx is can be accessed from our GitHub page https://github.com/HenriquesLab/NanoPyx. The Liquid Engine is available through its github repository page https://github.com/HenriquesLab/LiquidEngine/. These resources are fully open-source, providing users with tutorials, Jupyter Notebooks for Google Colab and many real-life example datasets for training and testing. |

For manuscripts utilizing custom algorithms or software that are central to the research but not yet described in published literature, software must be made available to editors and reviewers. We strongly encourage code deposition in a community repository (e.g. GitHub). See the Nature Portfolio guidelines for submitting code & software for further information.

## Data

Policy information about availability of data

All manuscripts must include a data availability statement. This statement should provide the following information, where applicable:
- Accession codes, unique identifiers, or web links for publicly available datasets
- A description of any restrictions on data availability
- For clinical datasets or third party data, please ensure that the statement adheres to our policy

> The example datasets we use to showcase NanoPyx are available for download in Zenodo (links provided in Supplementary Table S1 and our GitHub page).

## Human research participants

Policy information about studies involving human research participants and Sex and Gender in Research.

| | |
|---|---|
| Reporting on sex and gender | N/A |
| Population characteristics | N/A |
| Recruitment | N/A |
| Ethics oversight | N/A |

Note that full information on the approval of the study protocol must also be provided in the manuscript.

# Field-specific reporting

Please select the one below that is the best fit for your research. If you are not sure, read the appropriate sections before making your selection.

☒ Life sciences  ☐ Behavioural & social sciences  ☐ Ecological, evolutionary & environmental sciences

For a reference copy of the document with all sections, see nature.com/documents/nr-reporting-summary-flat.pdf

# Life sciences study design

All studies must disclose on these points even when the disclosure is negative.

| | |
|---|---|
| Sample size | We have used 2 biological datasets, described in the data availability section, and images simulated on the fly, available in the jupyter notebooks described in the code availability section, to generate the benchmarks of different methods |
| Data exclusions | No data was excluded |
| Replication | As images were only used as examples of the type of image that can be run in the methods no replication was needed. |
| Randomization | As images were only used as examples of the type of image that can be run in the methods no randomization was needed. |
| Blinding | As images were only used as examples of the type of image that can be run in the methods no blinding was needed. |

# Reporting for specific materials, systems and methods

We require information from authors about some types of materials, experimental systems and methods used in many studies. Here, indicate whether each material, system or method listed is relevant to your study. If you are not sure if a list item applies to your research, read the appropriate section before selecting a response.

## Materials & experimental systems

| n/a | Involved in the study |
|---|---|
| ☐ | ☒ Antibodies |
| ☐ | ☒ Eukaryotic cell lines |
| ☒ | ☐ Palaeontology and archaeology |
| ☒ | ☐ Animals and other organisms |
| ☒ | ☐ Clinical data |
| ☒ | ☐ Dual use research of concern |

## Methods

| n/a | Involved in the study |
|---|---|
| ☒ | ☐ ChIP-seq |
| ☒ | ☐ Flow cytometry |
| ☒ | ☐ MRI-based neuroimaging |

# Antibodies

| Antibodies used | Anti-α-Tubulin antibody, Mouse monoclonal, clone DM1A (Sigma, Catalog #T6199) (1:250); Anti-Tubulin-α Antibody, Mouse monoclonal, clone 10D8 (BioLegend, Catalog #625901) (1:500); Anti-Human Septin 7 IgG, Rabbit Polyclonal, (IBL, Catalog # JP18991) (1:100); Anti-Tubulin Beta 3 (TUBB3), Mouse monoclonal, clone AA10 (BioLegend, Catalog#657401) (1:500); Conjungated F(ab')2-Goat anti-Mouse IgG – Alexa Fluor 647 (ThermoFisher, Catalog #A-21237) (1:200); Conjugated Goat anti-Rabbit IgG - Alexa Fluor™ 555 (ThermoFisher Catalog #A-21428) (1:200). |
|---|---|
| Validation | Anti-α-Tubulin antibody (Catalog #T6199). Isotype: IgG1. Verified reactivity (VR): yeast, mouse, amphibian, human, rat, chicken, fungi, bovine. Antibody Type (AT): Monoclonal. Host species: Mouse. Concentration 1mg/ml. Application in immunoblotting, immunocytochemistry, immunofluorescence radioimmunoassay and western blot. Independent enhanced validation: antibody specificity demonstrated using multiple antibodies against target in immunohistochemistry or immunocytochemistry.<br>Anti-Tubulin-α Antibody (Catalog #625901). Isotype: Mouse IgM, κ. VR: Human, mouse, rat and all species. AT: monoclonal. Host species: Mouse. Concentration 0.5mg/ml. Application in western blotting (quality tested), immunohistochemistry – Paraffin and immunocytochemistry.<br>Anti-Human Septin 7 IgG (IBL, Catalog #JP18991). Isotype: IgG. VR: Human, mouse and rat. AT: Polyclonal. Host species: Rabbit. Concentration 0.1mg/ml. Application in western blotting, immunohistochemistry and immunoprecipitation.<br>Anti-Tubulin Beta 3 (TUBB3) (BioLegend, Catalog #657401). Isotype: Mouse IgG2a. VR: Mouse, Rat, Human. AT: Monoclonal. Host species: mouse. Concentration: 0.5mg/ml. Application in western blotting (quality tested), immunocytochemistry (verified), flow cytometry, immunofluorescence microscopy and spatial biology (IBEX). Knock-out validated.<br>Conjugated F(ab')2-goat anti-mouse IgG – Alexa Fluor 647 (ThermoFisher, Catalog #A-21237). Isotype: IgG. VR: Mouse. AT: Polyclonal. Host species: Goat/IgG. Concentration 2mg/ml. Application in western blotting, immunohistochemistry and immunocytochemistry. Cross adsorbed: against human IgG and serum.<br>Conjugated Full antibody-Goat anti-Rabbit IgG - Alexa Fluor 555 (ThermoFisher Catalog #A-21428); Isotype: IgG. VR: Rabbit. AT: Polyclonal. Host species: Goat/IgG. Concentration: 2mg/ml. Application in immunohistochemistry, immunocytochemistry and flow cytometry. Cross adsorbed: against human IgG, human serum, mouse IgG, mouse serum and bovine serum. |

# Eukaryotic cell lines

Policy information about cell lines and Sex and Gender in Research

| Cell line source(s) | A549 cell line (The European Coollection of Authenticated Cell Cultures (from ECACC Catalog # 86012804); Human Umbilical Vein Endothelial Cells (HUVEC) (from PromoCell C-12203) |
|---|---|
| Authentication | Cell lines were not authenticated |
| Mycoplasma contamination | Cells lines used tested negative for Mycoplasma contamination |
| Commonly misidentified lines (See ICLAC register) | None |

