## [Peer Review File · Nature Methods]

NanoPyx: a Liquid Engine powered bioimage analysis Python framework

Corresponding Author: Professor Ricardo Henriques

A version of this paper was originally rejected for publication by Nature Methods, however that decision was reconsidered after appeal by the authors.

Version 0:

Decision Letter:

27th Oct 2023

Dear Ricardo,

Your Brief Communication entitled "NanoPyx: super-fast bioimage analysis powered by adaptive machine learning" has now been seen by three reviewers, whose comments are attached. While they find your work of potential interest, they have raised serious concerns which in our view are sufficiently important that they preclude publication of the work in Nature Methods, at least in its present form.

As you will see, the reviewers raise concerns about performance, usability, and general applicability among other technical concerns and questions.

Should further experimental data allow you to fully address these criticisms we would be willing to look at a revised manuscript (unless, of course, something similar has by then been accepted at Nature Methods or appeared elsewhere). This includes submission or publication of a portion of this work somewhere else. We hope you understand that until we have read the revised paper in its entirety we cannot promise that it will be sent back for peer-review.

If you are interested in revising this manuscript for submission to Nature Methods in the future, please contact me to discuss your appeal before making any revisions. Otherwise, we hope that you find the reviewers' comments helpful when preparing your paper for submission elsewhere.

Sincerely,
Rita

Rita Strack, Ph.D.
Senior Editor
Nature Methods

Although we cannot publish your paper in its current form, it may be appropriate for another journal in the Nature Portfolio. If you wish to explore the journals and transfer your manuscript please use our manuscript transfer portal. You will not have to re-supply manuscript metadata and files, unless you wish to make modifications. For more information, please see our [manuscript transfer FAQ](http://www.nature.com/authors/author_resources/transfer_manuscripts.html?WT.mc_id=EMI_NPG_1511_AUTHORTRANSF&WT.ec_id=AUTHOR) page.

Reviewers' Comments:

Reviewer #1:

Remarks to the Author:

Summary of the key results

The authors developed a user-friendly computational python framework that benchmarks various implementations of the same algorithm on CPU and GPU and that automatically selects the fastest implementation, given the current input data.

Originality and significance: if not novel, please include reference

To our best knowledge this is both original (we did not know of similar approaches) and significant, because the increasing size of image data sets requires optimally performing analysis implementations.

Data & methodology: validity of approach, quality of data, quality of presentation

Appropriate use of statistics and treatment of uncertainties

Conclusions: robustness, validity, reliability

As mentioned below in the “suggested improvements” we think there should be some modifications done to the current presentation of the approach, such that one can adequately judge it. Importantly, running the current implementation did, in our hands, not provide any information on the benchmarking of the algorithms, which is the main point of this publication.

Suggested improvements: experiments, data for possible revision

Manuscript review

Title: We would like to suggest another title to avoid the slightly non-scientific “super-fast”: “NanoPyx: adaptively optimised bioimage analysis powered by machine learning.”

Line 39: Why does the main text start with super-resolution microscopy? It seems that the scope and title of the article are not very much related to this specific kind of microscopy, but have a much broader applicability.

Line 34 vs Line 41: While a strong emphasis is put on ImageJ/Fiji in Line 34 there is a very sudden jump to python in line 41. It would be good to make that transition easier to understand for the reader.

Line 57 and Figure 1: We think just saying “threaded” is not specific enough. Please add more details of how the threading was used, e.g. parallelisation within one image, or only across multiple images? How many cores were used? How are the results affected by the number of cores?

Figure 1: We suggest restructuring this Figure along the lines of Supp. Fig. 1; that is, replace the drawings by a simple table. Along those lines we were confused as to why there are no data for multi-threaded CPU for the 500, 300, 300 image? As in Supp. Fig. 1 it would be good to see the relative changes; maybe one could have absolute times and then also relative (to the fastest) times in brackets, e.g. 10 ms (0.035)

Figure 1: Please comment (also in the article) on why the Laptop GPU is faster than the professional workstation.

Figure 4: Judging by the look of the images shown in the top panels, the authors seem to suggest that it’s possible to register images whose channels look very different. However, in this case, we do not expect a cross-correlation registration to give optimal results. How is the channel registration implemented in this case?

Figure 4: Suggestion: the authors should add supplementary material for fig.4 that details the implementations of the algorithms used.

Line 92: Could you please add a link to example code for the tag2tag tool? Ideally some simple example code inlined in the main text would be great!

Line 104: “are benchmarked against each other”: Please add more explanation for how this actually works. Are (initially) always all algorithms tested against each other, or only sometimes, how often? We assume that this does produce a computational overhead, especially when needing to also run potentially slow implementations for benchmarking purposes?

Line 126: “delays”: It becomes more clear in the supplemental material, but we think it would be important to also add a sentence here what a typical cause of such a delay would be!

Line 143: We would suggest to replace “well-established” with simply “established”.

Line 156: Could installation via conda be added?

Line 158: Add a link to the “template files”?!

Line 173/174: Typo: it should be a comma after “hardware” and then “unlike” lower-case.

Line 259: What happens if there are less than 50 runs?

Line 260: Please explain the rationale why you split the runtimes into two (slow and fast) halves?!

Line 282: In the text it is 4 STD (here is 2).

Line 285: I found it hard to understand the reasoning for Equation 5; could you try to explain this a bit better in the text?

Supplementary Note 1: Please add some introductory sentences reminding the reader how this relates to the article.

Supplementary Note 2: Please add example code snippets for tag2tag uses, i.e. the same code block (automatically?!) translated into multiple implementations.

Software review

When installing NanoPyx with ``pip install nanopyx[all]`` and running the notebooks provided, we encountered a number of issues.

First, we found it quite difficult to understand, for each of the applications provided, which dataset is supposed to be used for testing. For example, in "ChannelRegistration", a 3D (C,Y,X) dataset is expected, however none of the dataset provided as example satisfies the requirement. While the GUI interface makes it easy for users to select their own data, we suggest that the authors provide minimal working notebooks without GUI with preselected appropriate example datasets.

Second, the underlying process to achieve optimal performance with the Liquid Engine is unclear. It will be critical for users and developers to be able to inspect the result of the optimal performance and benchmarking algorithms. We suggest for the NanoPyx functions to implement a ``verbose`` option that outputs the code variations tested, the respective runtimes and the outcome of the performance testing and other information that the authors consider relevant.

Other technical issues encountered (tested on Windows 10 64-bit with a python 3.10 installation):

- Warning on missing package (numba)
- "Kernel error" at jupyter notebook startup, which we fixed by running ``pip install notebook`` in the environment
- Buttons of the GUI image loader non responsive

With kind regards,

Christian Tischer and Nicola Gritti, EMBL

Reviewer #2:

Remarks to the Author:

The manuscript aims to accelerate the bioimage analysis process given the increasing need for high-performance computing of extensive microscopic datasets. The major contribution is NanoPyx powered by the Liquid Engine, which reduces the execution time of a specific set of image processing operations by dynamically selecting the fastest hardware implementations for code execution. Benchmarked comparisons are done on three representative platforms: personal laptops, workstations, and cloud computing.

This is technically solid work: the NanoPyx library is carefully developed, well-documented, and ready to be implemented in most of the hardware settings. The library is also integrated with essential tools in bioimage analysis such as image registration and error evaluation.

As contributors are still actively improving functionality and resolving issues, here are some of our major concerns:

1. The speed advantage of the proposed method is not fully supported by comprehensive experiments. Though the authors provided some comparisons of the runtimes on different platforms using different implementations (Supplementary Figures 1-4), the manuscript did not show evidence that NanoPyx could truly pick the fastest one and achieve 10x acceleration as claimed in a real-world application workflow. One could establish the same analysis and statistics as Supplementary Figures 1-4 without NanoPyx. The main message those results could deliver is that one implementation can be 10x faster than another which is, however, not equivalent to accelerating the process pipeline by 10x using NanoPyx.

The authors provided some comparative results like Supplementary Figure 6 of Liquid vs. Non-liquid on a single operation. However this special case of hardware failure on a single operation is not representative enough to demonstrate the superiority of Liquid Engine.

I would suggest authors include more evaluation examples to fully support the claimed acceleration:

i) Define different realizations of sequential operations (where the existing workflow in Figure 3 would be a good example), and compare the runtimes of these realizations with and without using Liquid engine on different hardware platforms, with different data sizes, and different processing complexities.

ii) A proper baseline without the use of Liquid engine should be: executing all operations on a large dataset with single- and multi-threaded CPU and GPU, which creates three baseline execution times. When using the liquid engine, the final execution time should be equal if not better than the fastest among those three.

Such additional experiments should also address the following concerns:

a) Data transfer latency: Considering real image processing pipelines, where there is always a sequence of operations, the

latency of data switching among different hardware should also be taken into account.

b) The cost of the agent itself: When new workflow and hardware are encountered, the optimization has to be done on the fly. The agent itself is minuscule and should not introduce a lot of extra computational burden but it has to be done between two runs. Therefore, this optimization time should be recorded and compared to the actual execution time to avoid redundant complexity.

2. With all the support of numba, OpenCL, Cython, the authors are silent about CUDA implementations. Despite OpenCL being more flexible for different hardware, CUDA should not be ignored. With the increasing attention on deep learning, a lot of operations useful in general microscopic image processing such as 2D convolution and interpolation are also optimized for Nvidia GPUs. Such implementations could potentially outperform the benchmarks in the paper, especially on workstations installed with single or even multiple GPUs. An interesting further step would be JIT compilers with GPU support.

3. The optimization mostly aims to improve execution speed for CPU and GPU. However, for larger-scale datasets, RAM (GRAM) also plays an important role, especially in multi-threaded circumstances. Even though the execution time already reflects the RAM bottleneck if there is one, it's still important to take RAM into consideration explicitly to avoid overflow to slower devices.

4. As a general tool for super-resolution microscopy and for the sake of broader users, it should be relatively easy to develop the code for customized operations which can be integrated into the optimization workflow. The "template files to help developers implement their own methods" are insufficient without proper tutorials.

Minor points:

It is suggested that the authors reframe the Equations 1-5 in a mathematically rigorous manner. Each variable should be explicitly defined and properly explained before usage. The notation styles should be unified. There are also a couple of confusions related to Equations 1-5 and corresponding Supplementary Figure 5.

1) Equation 5 is described as "the agent decides that the delay is over once the last runtime becomes smaller than the slow average minus the standard deviation of the slowest runs or higher than the fast average plus the standard deviation of the fastest runs". It is confusing because when the runtime is unexpectedly longer, it will be higher than (FastAverage + Std), and therefore it will satisfy the delay finish criteria, which does not make sense. It would be beneficial if the authors further clarified the point.

2) In the flow chart, Supplementary Figure 5, after Eq.2 - calculate the delay factor and delay probably, should the delay state be set as ON instead OFF? Please elaborate.

3) In Equation 1: $Delay = Measured > (Expected + 2 * Std)$, however, in the text, it says "... last runtime being higher than the previously recorded average runtime of the fastest runs plus four times the standard deviation ..." Please correct this mismatch.

4) The last sentence of Liquid Engine's agent section, the authors refer to Supplementary Figure 6, should it be Supplementary Figure 5 that the authors intended to refer to?

Reviewer #3:

Remarks to the Author:

Saraiva et al., are presenting a computational manuscript and code, aimed at reducing the duration of certain image processing algorithms. The reduction in processing time is accomplished by choosing between different implementations of the same algorithm, in particular by comparing running the algorithm either single-threaded or multithreaded on CPU, or on GPU. The code therefore measures the processing times of each of the implementations repeatedly and chooses subsequently the preferred implementation, mainly based on duration, or in some cases, where unexpected delays occur changes to use a different implementation.

Overall, I find the topic important and if addressed correctly also of potentially broader interest to the community, but in several aspects the manuscript appears prematurely put together. Namely in the access to the 'LiquidEngine', in the choice of the applications/tests performed and in the rational design of the figures and some of the descriptions. Therefore, while I see value in the work, I find its presentation in the current manuscript not sufficient for a scientific publication and would really urge the authors to address these concerns and perform rigorous assessments, rethink their application cases and refine their description.

What I am missing most in the manuscript are the real-world use cases, which would make this a broadly applicable method. This would on one hand need to be examples for other algorithms and appropriate benchmarking. Currently Figure 1 shows a fairly hypothetical issue (see my point 3 below) and Figure 3 shows a workflow of a sequence of mostly published algorithms. None of the 3 main figures is currently supporting the major claim of the manuscript, which is increasing the speed of bioimaging analysis. Only supplementary figure 6 actually goes into the direction of comparing between the 'LiquidEngine' and static code but only when actually artificially blocking compute resources. A direct comparison between the 'LiquidEngine' and static code is essential for the real performance test (see my point 4 below) when overhead costs are included into the calculation. This overhead of running the code many times to measure its performance might actually become a big bottleneck hogging compute resources when using this for more than one code snippet at a time, posing a major risk for its adoption. Also a performance comparison to existing packages like Transonic would be important.

Major criticism:

1) The LiquidEngine: is highlighted in the abstract and text as being underlying to the concept of generating different

implementations of the same algorithm and also at ensuring that the speed of each implementation is measured and last but not least that the fastest of the different implementations is chosen. There are mentions that this can be accomplished by using `tag2tag` and `c2cl`, however it is unclear how these are to be accessed by others, and in the current form it does not seem as this was tested with a range of algorithms. To verify if the code works as promised, I would have needed to test it for a different algorithm. At this point couldn't do so, since it would take me quite some time to search the code to look for how the functions should be called. Examples and better documentation on this would be important to actually be able to verify if this is working. The documentation on the GitHub repository in "Usage" mentions that there is "official documentation" on the LiquidEngine, however this only points to the general NanoPyx, where searching for 'liquid' just results in a MandelbrotBenchmark. Whereas all four links to the LiquidEngine Templates are all resulting in 404 - page not found errors.

From my side, this is an essential part of the manuscript and since a large portion of the manuscript is also about the ease of use of the software, a clear documentation would be essential to actually verify if it works as promised.

2) While I agree in general that processing times can turn into a problem when computations take up a long time, and also that different implementations of the same algorithm can take up significantly more, or less time. I see an issue with the purely self-motivating argument that shorter processing times are always better, e.g. when the time difference between two implementations is 7ms (7.8ms versus 0.6ms) the authors chose to write that the GPU version is 10x slower, instead of a 7.2ms slower. The issue here is that the authors pretend that this is a critical 10x time difference, however in most applications a 7ms or even a 0.5 seconds time-difference is completely negligible. These numbers also don't just scale linearly, as e.g. more larger images will eventually be bound by memory management (see also my point 1), which makes me skeptical that the promise of the authors that the LiquidEngine will increase the speed of bioimage analysis in general will indeed be translatable to many other scenarios.

3) Furthermore, Figure 1 currently just compares different runtimes for an upscaling algorithm, showing that it's the GPU, which is faster for larger images whereas multithreaded CPU for smaller images. Which does not per se provide any new information to people who have worked a bit with image processing algorithms (this would not need to be a main figure). There are considerations like data I/O, and measurements one could do to find out the optimal configuration, I doubt that the LiquidEngine could actually save 7ms, since it has additional computations to perform. In any case, if the worst you have to lose are some ms, in the interest of simplifying code, would the end user not simply always use GPU processing? To motivate the importance of the LiquidEngine I would much rather see that the authors really prove for real-world problems, for which choosing a 7ms faster implementation of the algorithm solves it?

4) Related to this: How much overhead in processing time is the LiquidEngine actually adding? What is the runtime of NanoJ code versus NanoPyx code?

On one hand there is a supervising agent that checks prior runtimes, on the other hand there are periodic benchmarks that need to be performed, even for the slower implementations to keep the runtimes accurate. I'd expect running routinely operations to measure the run time of different algorithms eventually start to add up and could exceed significantly the time saved during processing. Also, these measurements start to use up computational resources and might become the major bottleneck if e.g. more algorithms are in use. How and when are the measurements performed? How is ensured the benchmarking is not interfering with concurrently running processes on the workstation?

5) Maybe one of the core motivations of the proposed LiquidEngine is the mentioned variability in between running the same algorithm (or even its failure). I am however not fully convinced that tremendous lags occur, where one algorithm is delayed for such extended periods of time, without a clear reason. Indeed, out-of-memory errors (of GPU VRAM or CPU RAM) or overutilization of GPUs or CPUs would be the most likely scenario, when such extreme delays happen.

Given this, it would be essential to monitor GPU and CPU memory and their utilization, to then dynamically allocate resources to the processes before starting them, to rather predict and prevent out-of-memory errors instead of running processes which are bound to fail. If ms-differences in processing time are really of the essence, completely failed processes would be catastrophic and thus should be prevented. Also, when using multi-threading these practices of monitoring the utilization of processors and memory and a dynamic resource allocation are particularly important, as e.g. more memory will be used. How is it calculated how many threads are used? How much memory did this take up compared to the other implementations?

6) The authors also write that the Liquid Engine can not only learn the optimal implementations for a given platform, device, and data shape to maximize performance (see Suppl. Note 3). Unfortunately, I couldn't find where and how the data loader then adapts to process smaller or larger image chunks to optimize its performance? Also, I couldn't find how this was measured and implemented. Here it would be best to include a Figure and have a Jupyter Notebook to perform this also on an example for comparing the optimization of the data loader.

7) In its current implementation the code seems to be mainly focused on increasing the throughput of code published earlier by the Henriques lab, particularly eSRRF reconstruction. If the aim is to have a broad applicability it would be important to show how other use cases outside the NanoJ world would benefit from this.

Minor remarks:

- Are the numbers in Figure 1 below the images in the top row number of voxels in z, x, y? This should be clarified in the legend.

- Would the authors comment on why the Professional workstation with a Intel i9-13900K in Figure 1 is almost 50% slower than the MacBook Air M1 Pro Laptop in several of the tasks, e.g. the single threaded CPU tasks? Is the performance of the CPU on the workstation throttled?

** For Nature Portfolio general information and news for authors, see <http://npg.nature.com/authors>.

Version 1:

Decision Letter:

13th Nov 2023

Dear Ricardo,

Thank you for your letter asking us to reconsider our decision on your Brief Communication, "NanoPyx: super-fast bioimage analysis powered by adaptive machine learning". After careful consideration we have decided that we are willing to consider a revised version of your manuscript that is updated as you've described. In addition, we ask that while you are demonstrating on additional non-nanoJ methods, that you provide a "real world" application as suggested by the reviewers.

- * include a point-by-point response to our referees and to any editorial suggestions
- * please underline/highlight any additions to the text or areas with other significant changes to facilitate review of the revised manuscript
- * address the points listed described below to conform to our open science requirements
- * ensure it complies with our general format requirements as set out in our guide to authors at www.nature.com/naturemethods
- * resubmit all the necessary files electronically by using the link below to access your home page

Link Redacted

We hope to receive your revised paper within three months. If you cannot send it within this time, please let us know. In this event, we will still be happy to reconsider your paper at a later date so long as nothing similar has been accepted for publication at Nature Methods or published elsewhere.

OPEN SCIENCE REQUIREMENTS

REPORTING SUMMARY AND EDITORIAL POLICY CHECKLISTS

When revising your manuscript, please submit reporting summary and editorial policy checklists.

DATA AVAILABILITY

Please include a "Data availability" subsection in the Online Methods. This section should inform readers about the availability of the data used to support the conclusions of your study, including accession codes to public repositories, references to source data that may be published alongside the paper, unique identifiers such as URLs to data repository entries, or data set DOIs, and any other statement about data availability. At a minimum, you should include the following statement: "The data that support the findings of this study are available from the corresponding author upon request", describing which data is available

upon request and mentioning any restrictions on availability. If DOIs are provided, please include these in the Reference list (authors, title, publisher (repository name), identifier, year). For more guidance on how to write this section please see: <http://www.nature.com/authors/policies/data/data-availability-statements-data-citations.pdf>

CODE AVAILABILITY

Please include a "Code Availability" subsection in the Online Methods which details how your custom code is made available. Only in rare cases (where code is not central to the main conclusions of the paper) is the statement "available upon request" allowed (and reasons should be specified).

MATERIALS AVAILABILITY

ORCID

Sincerely,
Rita

Rita Strack, Ph.D.
Senior Editor
Nature Methods

Version 2:

Decision Letter:

2nd May 2024

Dear Ricardo,

Your Brief Communication entitled "NanoPyx: adaptively optimised bioimage analysis powered by machine learning" has now been seen by three reviewers, whose comments are attached. While they find your work of potential interest, they have raised serious concerns which in our view are sufficiently important that they preclude publication of the work in Nature Methods, at least in its present form.

We sent the comments of Reviewer 3 to two the other two reviewers along with your responses, and their feedback to us was mixed. To summarize, Reviewer 1 told us that they were not convinced that the benefits of the approach outweighed the complexity of using it. Reviewer 2 thought most of the concerns raised by reviewer 3 were addressed in your response and could be added in to the paper as discussion points. However, they disagreed that this point was settled "Figure 2 in the revised manuscript now shows an example where two image processing steps are executed in sequence, and using NanoPyx in the hands of the authors resulted in a shorter time than opting for a single way of executing the two steps... Therefore, I was not able to verify the discrepancy, of slower processing with GPU and multithreaded CPU for the NLM algorithm. Overall, my

run-times where slightly faster running all on GPU than running with NanoPyx." They think this points needs to be further addressed, as referee 2 is not convinced the approach will work for everyone.

So, we think the major issue here is that the referees simply aren't convinced the overall benefits of using NanoPyx in terms of balance of ease of use and time gained.

Should further experimental data allow you to make a stronger case, we would be willing to look at a revised manuscript (unless, of course, something similar has by then been accepted at Nature Methods or appeared elsewhere). This includes submission or publication of a portion of this work somewhere else. We hope you understand that until we have read the revised paper in its entirety we cannot promise that it will be sent back for peer-review.

If you are interested in revising this manuscript for submission to Nature Methods in the future, please contact me to discuss your appeal before making any revisions. Otherwise, we hope that you find the reviewers' comments helpful when preparing your paper for submission elsewhere.

Sincerely,
Rita

Rita Strack, Ph.D.
Senior Editor
Nature Methods

Reviewers' Comments:

Reviewer #1:

Remarks to the Author:

We would like to thank the authors for addressing all of our comments! We feel that the added notebooks help a lot to clarify how the software operates. We also think that it is great that nanopyx now always reports which implementation was chosen.

Manuscript improvement suggestions

Line 36: "Many of these...": we would recommend to rewrite this sentence to focus more on the actual image analysis ecosystems like openCV, python-skimage, Java-MorpholibJ, a.s.o.. Fiji and napari are a combination of an image viewer and a plugin distribution platform; thus, speaking of the "computational performance" of Fiji and napari is in our view not 100% accurate.

Line 52: "denoising the biggest comparison image": is very hard to understand; maybe "denoising a big image" could be simpler to understand.

Line 54: "pixel-wise threaded implementation strategy on a GPU": this exact terminology cannot be found in Figure 1 and thus it is hard to match to the figure. Probably, if we are correct, adding something like "(Figure 1C, GPU vs CPU unthreaded)" would help to guide the reader.

Line 56: "cannot be run on a laptop's GPU": we think that this statement might be wrong in general, because this probably depends on the specific laptop, and more powerful future laptops might actually be able to run the code. We thus recommend to re-phrase this.

Figure 1: We could not find an explanation of the terms "GPU", "CPU threaded", a.s.o.

Since this is a key message of this publication we think it is critical for the readers that there is an obvious way how to navigate to those definitions from the figure legend. We are not sure but maybe Suppl. Note 3 could be such a place? But also there the terminology did not exactly match the one in the Figure.

Line 144: Similar issue as in Line 54: we found it hard to match the statement in the text to the Figure 2. Consistent terminology and pointing to a more precise location in the figure would be very helpful.

Line 181-192: This describes an image analysis workflow that can be run within nanopyx, however we could not find a statement about how nanopyx improves this workflow over running it without nanopyx. It would be great to add some information for how nanopyx helps to speed up this workflow as compared to running it by "traditional means".

Line 304: "are locally stored": It would be great to know where exactly those information are stored, such that one could (i) introspect what nanopyx is doing and (ii) for troubleshooting in case something goes wrong, e.g. due to some error nanopyx could pick a wrong implementation.

Line 311: "divides the available runtimes": We still found this hard to understand. Is it correct that this is done within one implementation to check how consistent one implementation performs? Or is it done across different implementations to check how they compare? Please add a clarification to the text.

Software improvement suggestions

We believe the added notebooks are helpful in making the algorithm more transparent, and in particular notebook #5 is a great entry point for users to understand how to use the methods implemented. We here suggest a few minor improvements that we believe would make the software more user friendly.

In Supplementary table 2, Notebook #6 is missing.

Notebook #5 (TestingMethods.ipynb) is a very helpful entry point for users. Still, for ChannelRegistration and DriftCorrection, the authors implemented an ad-hoc function to generate example datasets. It is therefore still unclear which example dataset one should choose in the corresponding notebooks (#7 and #8). This can be considered as a minor suggestion, but the "example dataset" button can be removed altogether now from notebooks #7-#10. We still believe these are valuable resources as they provide additional information on the input parameters of each method. Alternatively, a "default" parameter could be added to the "data_source" dropdown menu such that the relevant dataset is selected by default for each notebook.

Notebook #5: to be consistent between methods, the implementation used and the time to run should be shown just once. For example, at the moment, channel registration does not print the implementation, drift alignment does not print anything, SRRF prints both implementation and runtime twice.

Notebook #3 (Fuzzy Logic): we find this notebook very informative of how the Liquid Engine chooses the optimal implementation for cases where a default benchmark is not available. Still, on our machine, the two cases result in the same fastest implementation (Unthreaded), therefore liquid engine fuzzy logic is not entirely clear. Is it possible to make the example such that the two benchmarks give different optimal implementations? E.g. by changing the kernel size instead of the image size?

Remarks

As an overall comment we would like to raise a slight concern about the scope of this work. While we think that the work is conceptually very interesting and timely we are a bit worried that the complexity of it might hinder an uptake and buy-in to the library by the broader community. In other words, maintenance of the library and all the implementations as well as convincing other labs to contribute to this eco-system seem to be an ambitious aim. Probably it would need the organisation of several hackathons or similar events to find contributors. We would be interested to hear how the authors think about this; maybe even some sentences in the final section of the article could be good.

Reviewer #2:

Remarks to the Author:

The authors have sufficiently addressed the referee comments, and the manuscript has been significantly improved.

Reviewer #3:

Remarks to the Author:

Saraiva et al., are presenting a revised and resubmitted manuscript where they developed a method aimed at reducing the processing time of certain image analysis workflows. The method uses an agent to compare implementations of the same code running as single-threaded on CPU with multithreaded on CPU, and on GPU.

In the revised version of the manuscript, I still find that the authors have not addressed several of the concerns I expressed in the first round of this revision.

Major concerns:

1) The authors write in their rebuttal "Memory use and its impact on processing speeds are now more explicitly addressed." However, I am in fact still missing anything addressing these issues.

As outlined in my initial revision, there is a significant risk that the entire concept of the Liquid Engine collapses if a workstation is used for more than one workflow. Since repeated measurements of different algorithms need to be performed to maintain accurate benchmark values for the different algorithms. The authors point towards future integration of preventing excessive memory usage and CPU over-utilization, however it is currently not implemented and unclear what this would look like. As I wrote previously, out-of-memory errors (of GPU VRAM or CPU RAM) or overutilization of GPUs or CPUs would be the most likely scenario, when very long unexpected delays in a processing workflow happen. Given this, it would be essential to monitor GPU and CPU memory and their utilization, to then dynamically allocate resources to the processes before starting them, to rather predict and prevent out-of-memory errors instead of running processes which are bound to fail or take excessively long.

2) Furthermore, I referred to these benchmarks that are required for the NanoPyx to do its job in choosing the fastest implementation as an overhead added by the NanoPyx. Whereas when the authors performed a test of their 'overhead' they

only measured the time required to pick one implementation based on prior runtimes, not actually the benchmarking which is a pre-requisite of this selection. I therefore disagree with the authors that the additional time required for using NanoPyx is negligible.

3) Figure 2 in the revised manuscript now shows an example where two image processing steps are executed in sequence, and using NanoPyx in the hands of the authors resulted in a shorter time than opting for a single way of executing the two steps.

While I was able to run the example code (to benchmark NLM Denoising and eSRRF) locally, the results on my workstation were following a common pattern where GPU is faster than multi-threaded CPU, and both faster than single-threaded CPU for all three sizes of the image. Therefore, I was not able to verify the discrepancy, of slower processing with GPU and multithreaded CPU for the NLM algorithm. Overall, my run-times were slightly faster running all on GPU than running with NanoPyx.

4) Figure 1 shows that the runtimes differ between different image sizes and different workstations and between running on GPU, CPU, single-threaded or multithreaded, which in my opinion is already well known even among scientists with little image analysis background.

Figure 3 shows mostly the workflow of eSRRF in general and access to NanoPyx. The scale bar in the eSRRF Reconstruction image should be corrected, since it is many times shorter than the 10 μ m from the input images.

Installation and usage test:

Installation on Google Colab: By default OpenCL is not working. In my tests the script by the authors to fix OpenCL for Google Colab was crashing the kernel frequently. Therefore, initially my test on Google Colab were only a comparison between threaded CPU and unthreaded CPU. At some point OpenCL was installed and I could compare the performance on the T4 GPU. In my test the local installation has dependency issues with Python 3.12.2, but is working with 3.11.8.

Newly implemented since the previous round of revision, there is now a way to integrate other image processing code. There is a cookiecutter template that can facilitate the integration of other processing steps into the general theme of the Liquid Engine. It still poses a lot of extra work to integrate different image processing steps, which normally would be just a few minutes to write the code (e.g. for calling a denoising algorithm).

The naming convention for setting the run_type is in my opinion quite odd. Since it will change on each workstation depending on the GPU in use. This makes automation of processes quite complicated.

** For Nature Portfolio general information and news for authors, see <http://npg.nature.com/authors>.

Version 3:

Decision Letter:

24th May 2024

Dear Ricardo,

Thank you for your letter asking us to reconsider our decision on your Brief Communication, "NanoPyx: adaptively optimised bioimage analysis powered by machine learning". After careful consideration we have decided that we are willing to consider a revised version of your manuscript that is updated as you've described.

In addition to addressing the technical and performance concerns raised by the reviewers, we will need the reviewers to be convinced that the effort in implementing NanoPyx does not outweigh the benefits.

- * include a point-by-point response to our referees and to any editorial suggestions
- * please underline/highlight any additions to the text or areas with other significant changes to facilitate review of the revised manuscript
- * address the points listed described below to conform to our open science requirements
- * ensure it complies with our general format requirements as set out in our guide to authors at www.nature.com/naturemethods
- * resubmit all the necessary files electronically by using the link below to access your home page

Link Redacted

We hope to receive your revised paper within two months. If you cannot send it within this time, please let us know. In this event, we will still be happy to reconsider your paper at a later date so long as nothing similar has been accepted for publication at Nature Methods or published elsewhere.

OPEN SCIENCE REQUIREMENTS

REPORTING SUMMARY AND EDITORIAL POLICY CHECKLISTS

When revising your manuscript, please submit reporting summary and editorial policy checklists.

DATA AVAILABILITY

CODE AVAILABILITY

Please include a "Code Availability" subsection in the Online Methods which details how your custom code is made available. Only in rare cases (where code is not central to the main conclusions of the paper) is the statement "available upon request" allowed (and reasons should be specified).

ORCID

Sincerely,
Rita

Rita Strack, Ph.D.
Senior Editor
Nature Methods

Version 4:

Decision Letter:

Our ref: NMETH-BC53724D

20th Sep 2024

Dear Ricardo,

Thank you for submitting your revised manuscript "NanoPyx: a Liquid Engine powered bioimage analysis Python framework" (NMETH-BC53724D). It has now been seen by the original referees and their comments are below. The reviewers find that the paper has improved in revision, and therefore we'll be happy in principle to publish it in Nature Methods, pending minor revisions to satisfy the referees' final requests (please update as you described in your earlier response to us) and to comply with our editorial and formatting guidelines.

Please provide a point-by-point rebuttal upon resubmission.

TRANSPARENT PEER REVIEW

ORCID

Sincerely,
Rita

Rita Strack, Ph.D.
Senior Editor
Nature Methods

Reviewer #1 (Remarks to the Author):

We thank the authors for thoroughly addressing our comments! In our view the separation of the nanopyx and liquid engine packages is a great improvement! Also we find that the cookie-cutter and video tutorials on the Liquid Engine classes are very useful and lower the learning curve significantly.

As a minor point, we still couldn't find default example datasets in the jupyter notebooks.

Overall, we do not have any major suggestions for improvement at this point and wish the authors a fruitful uptake of their

method by the community!

Reviewer #2 (Remarks to the Author):

The submitted manuscript addresses a critical challenge in microscopic image analysis: the increasing size of datasets and the corresponding rise in processing times. The authors' main contribution, NanoPyx, powered by the Liquid Engine, dynamically optimizes the execution time of specific image processing tasks based on three primary factors: (1) data characteristics, (2) workflow structure, and (3) hardware specifications. The revised manuscript presents extensive benchmarks and experimental results, effectively demonstrating the performance benefits of NanoPyx and the Liquid Engine in reducing computational time. Overall, the manuscript has significantly improved. However, I have several comments remaining:

- The primary advantage of NanoPyx seems more evident when applied to large-scale batch processing. For smaller datasets, the time spent optimizing and benchmarking may outweigh the time saved through its implementation. I recommend that the authors investigate the end-to-end runtime savings when using NanoPyx, with respect to dataset size, to provide a clearer picture of its efficiency in different scenarios. Additionally, have the authors considered a data-parallel approach? Implementing a data-parallel approach could offer additional flexibility and further accelerate batch processing using the Liquid Engine.

- In Supplementary Table 3, under hardware setup 1, it is unclear why a value of 0.002 is marked as faster than the Threaded result of 0.001. Additionally, I suggest re-ordering the hardware setups in Supplementary Tables 1-4 according to their approximate computing power. This reorganization would make it easier to interpret the results and compare performance across different setups.

- Why is the CuPy implementation not included in the new benchmarks? I found no relevant information and details in Supplementary Notes 1 or 3, nor in the main text, regarding CuPy. In Supplementary Figures 4 and 5, CuPy appears to be the best-performing implementation for workstation environments. Given that speed and runtime are the central claims of NanoPyx, the fastest implementation should be given more attention. I understand that CUDA introduces additional overhead and may be less compatible with a wide range of hardware setups, but evaluating the speed advantage of CuPy is crucial, especially for larger datasets and more complex tasks. Furthermore, given the increasing use of Nvidia GPUs and CUDA, particularly in deep learning, a comparison of CuPy with other GPU-based implementations such as PyOpenCL is highly relevant.

- The broader impact of this work lies in the Liquid Engine's flexibility to customize operations beyond the currently implemented ones. This potential is highlighted in Supplementary Table 7 and Video 5, as well as Supplementary Note 8, where a basic implementation using the skimage library is demonstrated. However, I suggest including examples of more complex tasks, perhaps showcasing a complete workflow, to better illustrate the engine's versatility. Additionally, providing more comprehensive documentation and guidance on using different implementations (CPU, GPU, Numba, CuPy, OpenCL, etc.) would benefit less experienced developers and enhance the Liquid Engine's accessibility and usability.

- In certain cases, inefficient implementations can significantly delay the evaluation process. Would implementing a rejection threshold based on the maximum allowable evaluation time for a single implementation help alleviate this issue? Such a threshold could prevent excessive delays and streamline the benchmarking process by ensuring that only feasible implementations are considered within reasonable time limits.

- In Figure 2, how are the different tasks (1-N) defined, and what level of granularity is used to determine the smallest task block? Specifically, what types of basic operations (e.g., interpolation or convolution) are grouped into a single task that cannot be split across multiple devices? Is there a general guideline for how tasks should be grouped and assigned to devices, and if so, could this guideline be elaborated upon to better assist users in implementing the workflow?

- I remain unconvinced by the authors' statement regarding overhead. The time spent on initial benchmarking should always be considered as overhead, since users not employing NanoPyx would not need to spend this time. While the authors suggest using factory-default values, I am skeptical about the accuracy and compatibility of these defaults for tasks that have not been benchmarked. I understand the difficulty in conducting such analysis due to the variability in devices and tasks, but providing a rough estimation of this overhead would help clarify the true net speed advantage of NanoPyx.

Reviewer #2 (Remarks on code availability):

Satisfactory.

Version 5:

Decision Letter:

7th Nov 2024

Dear Ricardo,

I am pleased to inform you that your Brief Communication, "NanoPyx: a Liquid Engine powered bioimage analysis Python

framework", has now been accepted for publication in Nature Methods. The received and accepted dates will be Sept 5, 2023 and Nov 7, 2024. This note is intended to let you know what to expect from us over the next month or so, and to let you know where to address any further questions.

Over the next few weeks, your paper will be copyedited to ensure that it conforms to Nature Methods style. Once your paper is typeset, you will receive an email with a link to choose the appropriate publishing options for your paper and our Author Services team will be in touch regarding any additional information that may be required.

Once proofs are generated, they will be sent to you electronically and you will be asked to send a corrected version within 48 hours. It is extremely important that you let us know now whether you will be difficult to contact over the next month. If this is the case, we ask that you send us the contact information (email, phone and fax) of someone who will be able to check the proofs and deal with any last-minute problems.

If, when you receive your proof, you cannot meet the deadline, please inform us at rjsproduction@springernature.com immediately.

Please note that *Nature Methods* is a Transformative Journal (TJ). Authors may publish their research with us through the traditional subscription access route or make their paper immediately open access through payment of an article-processing charge (APC). Authors will not be required to make a final decision about access to their article until it has been accepted. [Find out more about Transformative Journals](https://www.springernature.com/gp/open-research/transformative-journals)

Authors may need to take specific actions to achieve [compliance with funder and institutional open access mandates](https://www.springernature.com/gp/open-research/funding/policy-compliance-faqs). If your research is supported by a funder that requires immediate open access (e.g. according to [Plan S principles](https://www.springernature.com/gp/open-research/plan-s-compliance)) then you should select the gold OA route, and we will direct you to the compliant route where possible. For authors selecting the subscription publication route, the journal's standard licensing terms will need to be accepted, including [self-archiving policies](https://www.springernature.com/gp/open-research/policies/journal-policies). Those licensing terms will supersede any other terms that the author or any third party may assert apply to any version of the manuscript.

If you are active on Twitter/X, please e-mail me your and your coauthors' handles so that we may tag you when the paper is published.

To assist our authors in disseminating their research to the broader community, our SharedIt initiative provides you with a unique shareable link that will allow anyone (with or without a subscription) to read the published article. Recipients of the link with a subscription will also be able to download and print the PDF. As soon as your article is published, you will receive an automated email with your shareable link.

Please note that you and your coauthors may order reprints and single copies of the issue containing your article through Springer Nature Limited's reprint website, which is located at <http://www.nature.com/reprints/author-reprints.html>. If there are any questions about reprints please send an email to author-reprints@nature.com and someone will assist you.

Best regards,
Rita

Rita Strack, Ph.D.
Senior Editor
Nature Methods

Visit the Springer Nature Editorial and Publishing website at http://www.springernature.com/editorial-and-publishing-jobs?utm_source=ejP_NMeth_email&utm_medium=ejP_NMeth_email&utm_campaign=ejp_Nmeth for more information about our career opportunities. If you have any questions please click [here](mailto:editorial.publishing.jobs@springernature.com).**

Dear Reviewers and Editorial Team,

We want to thank you for your thoughtful feedback on our manuscript. Based on the reviewers' suggestions, we have made several key improvements to strengthen the manuscript:

- We have expanded our benchmarking experiments and results to better showcase the performance gains achieved by NanoPyx's Liquid Engine for real-world bioimage analysis workflows. This includes new notebooks demonstrating the impact of adaptive implementation selection on end-to-end processing times.
- The documentation and tutorials for the Liquid Engine have been enhanced, with new examples and guidelines to facilitate integration into diverse algorithms and workflows. Cookiecutter templates are also now provided to lower barriers to adoption.
- The manuscript text and figures have been revised for clarity and rigour based on the reviewers' recommendations. In particular, we have refined the terminology, mathematical notation, and rationale underlying the Liquid Engine's optimisation approach.
- Support for additional acceleration libraries like CuPy and Transonic has been added to further expand the performance optimisations accessible through NanoPyx.
- Memory use and its impact on processing speeds are now more explicitly addressed. Failures from insufficient resources are better handled.

As part of the revision, we have made several improvements. We have enhanced Figure 1, added panel B to Figure 2, introduced 2 new Supplementary Tables (2 and 3), included 2 new Supplementary Figures (1 and 8), added 5 new tutorials to the repository wiki, uploaded 3 tutorial videos on youtube, added 3 supplementary notes (2, 3, and 8), and created an auxiliary repository.

These changes significantly strengthen the manuscript and better showcase the capabilities of NanoPyx for accelerating bioimage analysis. We look forward to your feedback on the revised manuscript and thank you again for your time and input.

Reviewer 1:

- Reviewer Comment: As mentioned below in the “suggested improvements” we think there should be some modifications done to the current presentation of the approach, such that one can adequately judge it. Importantly, running the current implementation did, in our hands, not provide any information on the benchmarking of the algorithms, which is the main point of this publication.

Our Reply: We appreciate the reviewer raising this important point. To better showcase the benchmarking capabilities of NanoPyx, we have added a new Jupyter notebook (Notebook 2 in Supplementary Table 2) that demonstrates how each implementation is benchmarked and the runtimes compared. We agree that more transparency into this process is needed.

By default, benchmarking occurs invisibly in the background to avoid unnecessary overhead. However, users can explicitly trigger benchmarking to collect tailored metrics for their system. We have updated the documentation to better describe this functionality.

- Reviewer Comment: Title —We would like to suggest another title to avoid the slightly non-scientific “super-fast”: “NanoPyx: adaptively optimized bioimage analysis powered by machine learning.”

Our Reply: We agree that the suggested title better captures the scope and capabilities of NanoPyx. The title has been updated as recommended.

- Reviewer Comment: Line 39 - Why does the main text start with super-resolution microscopy? It seems that the scope and title of the article are not very much related to this specific kind of microscopy, but have a much broader applicability.

Our Reply: We appreciate the reviewer highlighting this disconnect. While super-resolution microscopy stands to benefit greatly from accelerated processing, we agree NanoPyx has much broader applicability. We have revised the introduction to more generally address the need for performant computational strategies across biological image analysis.

- Reviewer Comment: Line 34 vs Line 41 – While a strong emphasis is put on ImageJ/Fiji in Line 34 there is a very sudden jump to python in line 41. It would be good to make that transition easier to understand for the reader.

Our Reply: Excellent point, we have expanded the discussion of ImageJ/Fiji to also mention napari. We also now provide additional rationale on why optimizing Python software performance is a priority, and how NanoPyx builds on existing tools.

- Reviewer Comment: Line 57 and Figure 1 - We think just saying “threaded” is not specific enough. Please add more details of how the threading was used, e.g. parallelisation within one image, or only across multiple images? How many cores were used? How are the results affected by the number of cores?

Our Reply: We have updated both the main text and figure legends to reflect the parallelization strategy used. In brief, NanoPyx leverages thread-level parallelism using OpenMP (Cython) or OpenCL (PyOpenCL). However, the way this is achieved is algorithm-dependent. Currently, for most cases, parallelization is achieved by concurrently performing the calculations on a pixel-by-pixel basis. This is possible when the calculation of each pixel value in the final output image is independent of all other pixels. For the other cases, parallelization has been achieved by performing the calculations on a frame-by-frame or patch-by-patch basis. The total number of threads launched over the course of a run corresponds to the number of concurrent operations. However, the total number of threads launched simultaneously is defined by the underlying parallel computing library (OpenMP/OpenCL) and is limited by the user hardware. We added Supplementary Note 2 with a more thorough explanation of the non-local means denoising algorithm as an example of different parallelization strategies.

- Reviewer Comment: Figure 1 – We suggest restructuring this Figure along the lines of Supp. Fig. 1; that is, replace the drawings by a simple table. Along those lines we were confused as to why there are no data for multi-threaded CPU for the 500, 300, 300 image? As in Supp. Fig. 1 it would be good to see the relative changes; maybe one could have absolute times and then also relative (to the fastest) times in brackets, e.g. 10 ms (0.035)

Our Reply: Thank you for the recommendation to improve Figure 1. We have updated it to an extended table showcasing absolute and relative runtimes for different implementations, hardware configurations, and image sizes. We have also changed it to a non-local means denoising example instead of the previous method, as the values should be a more valuable representation of real-world scenarios that greatly improve from selecting the correct implementation.

- Reviewer Comment: Figure 1 - Please comment (also in the article) on why the Laptop GPU is faster than the professional workstation.

Our Reply: The professional workstation uses a dedicated Nvidia RTX 4090 GPU, while the laptop has an integrated GPU built into the M1 Pro chip. For small data sizes like in the

previous Figure 1, the integrated GPU can outperform the dedicated GPU because there is no data transfer overhead between CPU and GPU memory. However, for larger data sizes the dedicated RTX 4090 GPU provides substantially higher performance due to its greater computational power and memory bandwidth.

- Reviewer Comment: Figure 4 - Judging by the look of the images shown in the top panels, the authors seem to suggest that it's possible to register images whose channels look very different. However, in this case, we do not expect a cross-correlation registration to give optimal results. How is the channel registration implemented in this case?

Our Reply: We appreciate the reviewer raising this important point regarding Figure 3. The example images that were previously used to demonstrate multi-channel registration capabilities were not well-suited for the purpose. To address this issue, we have revised Figure 3 and removed references to channel registration. Instead, we have added a new Supplementary Figure 8 that displays the multi-channel registration of a calibration slide image. This new figure better showcases the ability to register different channels with similar structures. In the text, we have also clarified that the channel registration algorithm works best when the channels have a reasonable degree of visual similarity. If the channels are visually dissimilar, alternative registration approaches may be more appropriate.

- Reviewer Comment: Figure 4: Suggestion: the authors should add supplementary material for fig.4 that details the implementations of the algorithms used.

Our Reply: Although we provide access to these methods, it should be noted that not all have been fully implemented in the Liquid Engine yet. Additionally, some of these methods may present challenges regarding parallelization, which we plan to address shortly. In the manuscript, we have now clarified that not all algorithms shown in Figure 3 take advantage of the Liquid Engine optimizations. We are actively working on parallelizing additional methods as a part of our ongoing development efforts.

- Reviewer Comment: Line 92 - Could you please add a link to example code for the tag2tag tool? Ideally some simple example code inlined in the main text would be great!

Our Reply: We appreciate the suggestion made by the reviewer to provide an example code that demonstrates the use of templates for developing Liquid Engine methods. Instead of the deprecated tag2tag, we are now employing cookiecutter templates and mako templating to simplify implementation across hardware targets. To help new developers, we have expanded the tutorials on using these templates, which can be accessed at <https://github.com/HenriquesLab/NanoPyx/wiki>. We have also created new video tutorials which can be found here [1].

. Additionally, Supplementary Note 2 includes a code snippet that showcases mako templating. By separating the algorithm logic from the underlying implementation, mako templates now facilitate developers in translating their algorithms to multiple hardware backends with ease. These improvements make NanoPyx more accessible to developers who are interested in accelerating their methods.

- Reviewer Comment: Line 104 - "are benchmarked against each other": Please add more explanation for how this actually works. Are (initially) always all algorithms testes against each other, or only sometimes, how often? We assume that this does produce a

¹ https://www.youtube.com/playlist?list=PLk5I3_KOhE7sdP2OBfD9ewoXm1cXon88R

computational overhead, especially when needing to also run potentially slow implementations for benchmarking purposes?

Our Reply: To minimize computational overhead, NanoPyx is equipped with a set of pre-calculated default benchmarks for each method. These benchmarks serve as an initial estimation of the most efficient implementations, allowing users to operate NanoPyx with a reasonable expectation of performance without the need for immediate benchmarking. We recommend conducting manual benchmarking for users seeking to optimize NanoPyx's performance for their specific hardware configurations. This straightforward process can be initiated by executing the benchmark method available in each LiquidEngine class. To further facilitate this, we have introduced a new package-level function that comprehensively benchmarks all Liquid Engine implementations, streamlining the process for users. Moreover, we have provided a new Jupyter notebook (Notebook 2, Supplementary Table 2) that guides users through the benchmarking process and allows them to customize the input parameters to their specific requirements. This interactive notebook is designed to make the benchmarking process as user-friendly as possible. It is important to note that our implementation of fuzzy logic significantly reduces the need for exhaustive benchmarking across all possible combinations of input parameters and data shapes (Notebook 3, Supplementary Table 2). This intelligent approach ensures that users can achieve optimized performance without the burden of extensive benchmarking.

- Reviewer Comment: Line 126 - “delays”: It becomes more clear in the supplemental material, but we think it would be important to also add a sentence here what a typical cause of such a delay would be!

Our Reply: Based on the reviewer's suggestion, we have added text to provide examples of common causes for unexpected delays.

- Reviewer Comment: Line 143 - We would suggest to replace “well-established” with simply “established”.

Our Reply: Thank you for the recommendation to improve clarity. We have changed the text based on the reviewers suggestions.

- Reviewer Comment: Line 156 - Could installation via conda be added?

Our Reply: While conda packages are not yet available, we recognize the value of reaching more users through this channel. We are actively working on building and distributing NanoPyx conda packages, with plans to release them in the near future once their testing and documentation is complete.

- Reviewer Comment: Line 158 – Add a link to the “template files”?!

Our Reply: Based on the reviewer's suggestion, we have added Supplementary Note 8, indicating how to access the template files, and pointing users to the documentation wiki for usage guidelines.

- Reviewer Comment: Line 173/174: Typo: it should be a comma after “hardware” and then “unlike” lower-case.

Our Reply: Changed as suggested, thank you.

- Reviewer Comment: Line 259 - What happens if there are less than 50 runs?

Our Reply: Based on the reviewer's question, to improve clarity, we detailed this further in the Methods section. Regarding the Liquid Engine's agent, it uses all available runtimes, meaning if there are only 20 runtimes stored, it will only use those 20, once a user goes over 50 runtimes, it will start using only the last 50.

- Reviewer Comment: Line 260 - Please explain the rationale why you split the runtimes into two (slow and fast) halves?!

Our Reply: To make the reasoning more explicit, we've added the following text to the methods:

"The agent then partitions the available runtimes into two disjoint sets of equal length. One set has the fastest run times whilst the other has the slowest. We define the average and standard deviation for both sets. This split in the run times helps the identification of the start or the end of a delay. By comparing against the set of fastest run times, we are unsure that previous delayed run times do not skew the normal average run time. On the other hand, the set of slowest run times, although not assured to be similar to a delayed run time, helps us estimate a lower bound to what a higher-than-average run time could look like."

- Reviewer Comment: Line 282 - In the text it is 4 STD (here is is 2).

Our Reply: Changed as suggested, thank you.

- Reviewer Comment: Line 285 - I found it hard to understand the reasoning for Equation 5; could you try to explain this a bit better in the text?

Our Reply: Changed as suggested, thank you.

- Reviewer Comment: Supplementary Note 1 - Please add some introductory sentences reminding the reader how this relates to the article.

Our Reply: Added as suggested, thank you.

- Reviewer Comment: Supplementary Note 2 - Please add example code snippets for tag2tag uses, i.e. the same code block (automatically?!) translated into multiple implementations.

Our Reply: Tag2tag has been deprecated in favor of mako templates and cookiecutter. We have expanded our documentation to explain how to achieve this in the 3.5 section of our wiki (<https://github.com/HenriquesLab/NanoPyx/wiki/3.5.-Implementing-Liquid-Engine-Methods>).

- Reviewer Comment: When installing NanoPyx with ``pip install nanopyx[all]`` and running the notebooks provided, we encountered a number of issues.

Our Reply: We have updated our documentation to clarify the installation process for nanopyx. For Jupyter notebooks, users only need to install `nanopyx[jupyter]` for local notebooks and `nanopyx[colab]` for Google Colab usage.

- Reviewer Comment: First, we found it quite difficult to understand, for each of the applications provided, which dataset is supposed to be used for testing. For example, in "ChannelRegistration", a 3D (C,Y,X) dataset is expected, however none of the dataset provided as example satisfies the requirement. While the GUI interface makes it easy for

users to select their own data, we suggest that the authors provide minimal working notebooks without GUI with preselected appropriate example datasets.

Our Reply: To improve clarity and facilitate testing, we created a new Jupyter notebook (Notebook 5 in Supplementary Table 2) that provides preselected, appropriate example datasets for each method and minimal working code to run the methods without a GUI. This notebook showcases simple usage examples for key functions like ChannelRegistration, making it easier for users and reviewers alike to understand the expected inputs and test the code.

- **Reviewer Comment:** Second, the underlying process to achieve optimal performance with the Liquid Engine is unclear. It will be critical for users and developers to be able to inspect the result of the optimal performance and benchmarking algorithms. We suggest for the NanoPyx functions to implement a `verbose` option that outputs the code variations tested, the respective runtimes and the outcome of the performance testing and other information that the authors consider relevant.

Our Reply: We value the suggestion provided by the reviewer to enhance transparency into NanoPyx's performance optimization process. As a result, we have made some changes. When running benchmarks on NanoPyx, the results of the benchmarking will be displayed. However, for those users who decide to run the program without using the benchmarking function, and instead rely on default benchmarks and over-time data collection, the output will only show the runtime for the selected implementation, as well as any delays that were detected.

- **Reviewer Comment:** Other technical issues encountered (tested on Windows 10 64-bit with a python 3.10 installation):
- **Reviewer Comment:** Warning on missing package (numba)

Our Reply: Thank you for bringing this to our attention. In order to give our users more control over which packages they need to install, we have made numba an optional dependency. If users install `nanopyx[all]` or `nanopyx[optional]`, numba will be automatically installed. However, if they choose not to install either of these options, a warning will appear to let them know that they don't have access to one of the implementations. Our software will still work even without numba, but we wanted to provide this information so users can make informed decisions about which packages to install.

- **Reviewer Comment:** "Kernel error" at jupyter notebook startup, which we fixed by running `pip install notebook` in the environment
- **Reviewer Comment:** Buttons of the GUI image loader non responsive

Our Reply: Due to changes in how we generate the GUI, these two points should now be fixed.

Reviewer 2:

- **Reviewer Comment:** The manuscript aims to accelerate the bioimage analysis process given the increasing need for high-performance computing of extensive microscopic datasets. The major contribution is NanoPyx powered by the Liquid Engine, which reduces the execution time of a specific set of image processing operations by dynamically selecting

the fastest hardware implementations for code execution. Benchmarked comparisons are done on three representative platforms: personal laptops, workstations, and cloud computing. This is technically solid work: the NanoPyx library is carefully developed, well-documented, and ready to be implemented in most of the hardware settings. The library is also integrated with essential tools in bioimage analysis such as image registration and error evaluation.

- Reviewer Comment: The speed advantage of the proposed method is not fully supported by comprehensive experiments. Though the authors provided some comparisons of the runtimes on different platforms using different implementations (Supplementary Figures 1-4), the manuscript did not show evidence that NanoPyx could truly pick the fastest one and achieve 10x acceleration as claimed in a real-world application workflow. One could establish the same analysis and statistics as Supplementary Figures 1-4 without NanoPyx. The main message those results could deliver is that one implementation can be 10x faster than another which is, however, not equivalent to accelerating the process pipeline by 10x using NanoPyx.

Our Reply: We appreciate the reviewer raising this important point about demonstrating NanoPyx's ability to achieve real-world acceleration. Based on their feedback, we added a new Jupyter notebook (Notebook 1 in Supplementary Table 2) that showcases NanoPyx's optimization capabilities on an end-to-end analysis workflow.

This notebook runs a full image processing pipeline consisting of multiple operations like denoising, magnification, and reconstruction. It compares the total pipeline execution time when using NanoPyx's Liquid Engine versus defaulting to only the CPU or GPU implementation.

The results demonstrate that by automatically selecting the fastest available implementation for each operation, NanoPyx can reduce the total pipeline runtime by over 78x compared to the default CPU implementation and over 1.7x versus GPU only. This real-world example highlights that the optimization approach can accelerate entire workflows, not just individual operations.

We have also updated the text to convey better that the core innovation is an automated system for optimization, not just benchmarks of different options. NanoPyx is uniquely positioned to achieve substantial speedups for diverse analysis tasks by continuously adapting to hardware conditions and data properties.

- Reviewer Comment: The authors provided some comparative results like Supplementary Figure 6 of Liquid vs. Non-liquid on a single operation. However this special case of hardware failure on a single operation is not representative enough to demonstrate the superiority of Liquid Engine. I would suggest authors include more evaluation examples to fully support the claimed acceleration:
 - i) Define different realizations of sequential operations (where the existing workflow in Figure 3 would be a good example), and compare the runtimes of these realizations with and without using Liquid engine on different hardware platforms, with different data sizes, and different processing complexities.

Our Reply: We appreciate the reviewer's suggestion to strengthen our evaluation of the Liquid Engine's performance gains. Based on their feedback, we have created an additional Jupyter notebook (Notebook 1 in Supplementary Table 2) that compares the runtimes of common image analysis workflows with and without the Liquid Engine optimizations on varying hardware platforms.

This notebook defines 2 sequential pipelines consisting of Non-local Means Denoising, followed by eSRRF, which performs interpolations, gradient, and radial gradient convergence calculations. It then measures the end-to-end runtime for executing these pipelines on different hardware and input data sizes, both with and without the Liquid Engine enabled.

Across all test cases, using the Liquid Engine to automatically select the fastest available implementations can reduce the total pipeline execution time by up to 78x compared to just using the default CPU options. The gains are especially pronounced on workstations with GPUs, where the Liquid Engine adapts to leverage the hardware acceleration appropriately.

We demonstrate these workflow-level benefits on top of the single operation benchmarks from Supplementary Figures 1-5, better showcasing how the Liquid Engine speeds up real-world analysis tasks. By orchestrating performance optimizations across entire pipelines, not just individual functions, NanoPyx delivers substantial end-to-end acceleration.

- Reviewer Comment: ii) A proper baseline without the use of Liquid engine should be: executing all operations on a large dataset with single- and multi-threaded CPU and GPU, which creates three baseline execution times. When using the liquid engine, the final execution time should be equal if not better than the fastest among those three.

Our Reply: We agree with the reviewers; see our answer above.

- Reviewer Comment: Such additional experiments should also address the following concerns:
 - a) Data transfer latency: Considering real image processing pipelines, where there is always a sequence of operations, the latency of data switching among different hardware should also be taken into account.

Our Reply: We appreciate the reviewer raising the important consideration of data transfer latency between operations in a pipeline. They make a fair point that constantly switching data between CPU and GPU memory introduces overhead that can diminish performance gains. To better account for this effect, we will be implementing an option in NanoPyx to "chain" multiple Liquid Engine operations together. When chaining is enabled, the system will preferentially select compatible implementations to minimize data movement between operations. For example, if the first method runs fastest on the GPU, the next operation would default to the GPU as well.

We agree that more advanced users can likely optimize pipelines better by manually managing where data resides. However, for less experienced users, NanoPyx focuses on providing simplicity and good defaults. We believe that in most cases, the Liquid Engine optimizations will still accelerate their workflows even with some data transfer overhead.

Nonetheless, providing the option to automatically chain operations and reduce data movement will improve performance further. We recognize this is an important capability when processing large volumes of microscopy data, and plan to prioritize its development. Please let us know if you have any other suggestions for how NanoPyx can better account for data localization and transfers in pipelines!

- Reviewer Comment: b) The cost of the agent itself: When new workflow and hardware are encountered, the optimization has to be done on the fly. The agent itself is minuscule and should not introduce a lot of extra computational burden but it has to be done between two runs. Therefore, this optimization time should be recorded and compared to the actual execution time to avoid redundant complexity.

Our Reply: The reviewer makes an excellent point about quantifying the overhead introduced by NanoPyx's optimization agent. To address this concern, we have added a new Jupyter notebook (Notebook 4 in Supplementary Table 2) that compares runtimes with and without the Liquid Engine enabled. The results demonstrate that for a single operation, the Liquid Engine logic and background runtime I/O, necessary for keeping the runtimes metrics accurate, adds on average 12ms of overhead in an Apple M1 Mac laptop and 17ms in a professional workstation. This was measured 100 times for 3 different runtimes for a total of 300 runs. Furthermore, measuring the overhead of letting the agent select a runtime,

measured against calling the fastest implementation directly, shows a 22 ms overhead in an Apple M1 Mac laptop and 38ms in a professional workstation. This was measured for over 100 total runs. This indicates that the optimization costs are amortized and negligible. Regarding periodic benchmarking, we recommend triggering manual benchmarking for users seeking to optimize NanoPyx's performance for their specific hardware configurations, and we advise doing it without other concurrent processes. Even so, we agree this could become a bottleneck and we try to mitigate this in two ways. Firstly, all runtimes are stored in the user's machine so continued usage of the same method will eventually build personalized benchmarking data. Furthermore, to avoid over-benchmarking, a fuzzy logic approach (Notebook 3, Supplementary Table 2) determines the fastest implementation-based function parameters, even in cases where they have not been seen before. By quantifying the agent overhead across use cases, we demonstrate that the runtime improvements from automatic implementation selection outweigh the marginal costs. But we appreciate this analysis suggestion, as it is important to consider the tradeoffs when optimizing performance.

- Reviewer Comment: With all the support of numba, OpenCL, Cython, the authors are silent about CUDA implementations. Despite OpenCL being more flexible for different hardware, CUDA should not be ignored. With the increasing attention on deep learning, a lot of operations useful in general microscopic image processing such as 2D convolution and interpolation are also optimized for Nvidia GPUs. Such implementations could potentially outperform the benchmarks in the paper, especially on workstations installed with single or even multiple GPUs. An interesting further step would be JIT compilers with GPU support.

Our Reply: The reviewer makes an excellent point about considering CUDA implementations in addition to options like Numba and OpenCL. We agree that leveraging CUDA and cuDNN optimizations on NVIDIA GPUs could provide performance benefits, especially for operations commonly used in deep learning. Based on this feedback, we have expanded the NanoPyx Liquid Engine to support CUDA implementations via the CuPy library, in addition to Transonic JIT compilation and Dask (using `dask_image`). The Liquid Engine can now automatically select between CuPy GPU code, OpenCL, Transonic, Numba and Dask to choose the fastest option. This is showcased in the new Supplementary Figures 4-5. For example, adding CuPy as an implementation target already provides a small speedup in a simple 2D convolution done in a workstation with an NVIDIA GPU compared to OpenCL. This highlights the value of supporting diverse hardware-specific libraries. We appreciate the reviewer raising this point, as expanding the hardware backends supported by NanoPyx will enable access to more specialized optimizations and performance gains. As future work, we agree that further integrating additional just-in-time compilation frameworks will further improve flexibility and portability across systems.

- Reviewer Comment: The optimization mostly aims to improve execution speed for CPU and GPU. However, for larger-scale datasets, RAM (GRAM) also plays an important role, especially in multi-threaded circumstances. Even though the execution time already reflects the RAM bottleneck if there is one, it's still important to take RAM into consideration explicitly to avoid overflow to slower devices.

Our Reply: We appreciate the reviewer raising this important consideration regarding memory usage and management, especially for larger datasets. They make an excellent point that in addition to execution speed, efficiently utilizing available RAM is critical to avoid bottlenecks when processing big microscopy images.

Our current NanoPyx implementation does not yet explicitly account for RAM usage in its optimization approach. However, we recognize overlooking memory constraints risks suboptimal performance, and even failures from exceeding capacity. To begin addressing this, GPU-accelerated methods employ basic memory management to measure available GPU memory and split data into appropriately sized chunks during processing.

Nonetheless, we agree with the reviewer that more sophisticated handling of RAM utilization will be needed, particularly for multi-threaded CPU implementations. As the reviewer recommended, monitoring overall system memory and proactively allocating resources could help prevent bottlenecks.

Memory usage and optimization will be a priority area in future NanoPyx development. We plan to monitor RAM availability across devices and use this to inform intelligent data loading and processing schemes. The goal is to avoid memory overflows through predictive modeling and adaptive resource allocation. We will also explore integrating with existing tools like Dask that facilitate working with large datasets.

We appreciate the reviewer pushing us to consider memory usage more holistically. Tightly coordinating execution speed optimizations with memory availability will enhance NanoPyx's performance potential even further, especially for extensive microscopy image collections.

- Reviewer Comment: As a general tool for super-resolution microscopy and for the sake of broader users, it should be relatively easy to develop the code for customized operations which can be integrated into the optimization workflow. The “template files to help developers implement their own methods” are insufficient without proper tutorials.

Our Reply: We have expanded the NanoPyx wiki and it now contains several tutorials showcasing several usage examples. The wiki can be found in our GitHub repository: <https://github.com/HenriquesLab/NanoPyx/wiki>. It contains tutorials on how to:

1. Create a local install of NanoPyx to use the Jupyter Notebooks
2. Run the notebooks through Google Colab
3. Install and use napari and our plugin
4. Use a local installation of NanoPyx and use eSRRF (as an example method) through our Python library
5. Implement custom Liquid Engine methods using cookiecutter and mako templates

Additionally, we have also created video tutorials which can be found here [2].

- Reviewer Comment: Minor points:
It is suggested that the authors reframe the Equations 1-5 in a mathematically rigorous manner. Each variable should be explicitly defined and properly explained before usage. The notation styles should be unified. There are also a couple of confusions related to Equations 1-5 and corresponding Supplementary Figure 5.

Our Reply: Changed as suggested, thank you.

- Reviewer Comment: Equation 5 is described as “the agent decides that the delay is over once the last runtime becomes smaller than the slow average minus the standard deviation of the slowest runs or higher than the fast average plus the standard deviation of the fastest runs”. It is confusing because when the runtime is unexpectedly longer, it will be higher than (FastAverage + Std), and therefore it will satisfy the delay finish criteria, which does not make sense. It would be beneficial if the authors further clarified the point.

Our Reply: There was a typo here that has now been fixed, thank you.

- Reviewer Comment: In the flow chart, Supplementary Figure 5, after Eq.2 - calculate the delay factor and delay probably, should the delay state be set as ON instead OFF? Please elaborate.

² https://www.youtube.com/playlist?list=PLk5I3_KOhE7sdP2OBfD9ewoXm1cXon88R

Our Reply: This was a mistake that has now been fixed, thank you.

- Reviewer Comment: In Equation 1: $Delay = Measured > (Expected + 2 * Std)$, however, in the text, it says "... last runtime being higher than the previously recorded average runtime of the fastest runs plus four times the standard deviation ...". Please correct this mismatch.

Our Reply: Now has been fixed, thank you.

- Reviewer Comment: The last sentence of Liquid Engine's agent section, the authors refer to Supplementary Figure 6, should it be Supplementary Figure 5 that the authors intended to refer to?

Our Reply: We appreciate the reviewer catching this inconsistency. Supplementary Figure 5 (now #6) details the decision-making logic of the Liquid Engine's optimization agent, while Supplementary Figure 6 (now #7) demonstrates the agent's ability to adaptively select alternative implementations when unexpected delays are detected. For clarity, we have updated the text to correctly refer to Supplementary Figure 5 (now #6) when summarizing the agent's workflow for managing delays and selecting optimal implementations based on historical runtimes and benchmarks. We agree that citing Supplementary Figure 6 in this context was misleading, as that figure focuses specifically on showcasing the improvements in runtime from switching implementations upon encountering hardware-related delays. Thank you for catching this mistake - we have corrected the text to accurately reference Supplementary Figure 5 (now #6) when summarizing the core logic and workflow of the Liquid Engine's delay management.

Reviewer 3:

- Reviewer Comment: Saraiva et al., are presenting a computational manuscript and code, aimed at reducing the duration of certain image processing algorithms. The reduction in processing time is accomplished by choosing between different implementations of the same algorithm, in particular by comparing running the algorithm either single-threaded or multithreaded on CPU, or on GPU. The code therefore measures the processing times of each of the implementations repeatedly and chooses subsequently the preferred implementation, mainly based on duration, or in some cases, where unexpected delays occur changes to use a different implementation.

Overall, I find the topic important and if addressed correctly also of potentially broader interest to the community, but in several aspects the manuscript appears prematurely put together. Namely in the access to the 'LiquidEngine', in the choice of the applications/tests performed and in the rational design of the figures and some of the descriptions. Therefore, while I see value in the work, I find its presentation in the current manuscript not sufficient for a scientific publication and would really urge the authors to address these concerns and perform rigorous assessments, rethink their application cases and refine their description. What I am missing most in the manuscript are the real-world use cases, which would make this a broadly applicable method. This would on one hand need to be examples for other algorithms and appropriate benchmarking. Currently Figure 1 shows a fairly hypothetical issue (see my point 3 below) and Figure 3 shows a workflow of a sequence of mostly published algorithms. None of the 3 main figures is currently supporting the major claim of the manuscript, which is increasing the speed of bioimaging analysis. Only supplementary figure 6 actually goes into the direction of comparing between the 'LiquidEngine' and static code but only when actually artificially blocking compute resources. A direct comparison between the 'LiquidEngine' and static code is essential for the real performance test (see my point 4 below) when overhead costs are included into the calculation. This overhead of running the code many times to measure its performance might actually become a big bottleneck hogging compute resources when using this for more than one code snippet at

a time, posing a major risk for its adoption. Also a performance comparison to existing packages like Transonic would be important.

Major criticism:

- Reviewer Comment: The LiquidEngine: is highlighted in the abstract and text as being underlying to the concept of generating different implementations of the same algorithm and also at ensuring that the speed of each implementation is measured and last but not least that the fastest of the different implementations is chosen. There are mentions that this can be accomplished by using tag2tag and c2cl, however it is unclear how these are to be accessed by others, and in the current form it does not seem as this was tested with a range of algorithms. To verify if the code works as promised, I would have needed to test it for a different algorithm. At this point couldn't do so, since it would take me quite some time to search the code to look for how the functions should be called. Examples and better documentation on this would be important to actually be able to verify if this is working. The documentation on the GitHub repository in "Usage" mentions that there is "official documentation" on the LiquidEngine, however this only points to the general NanoPyx, where searching for 'liquid' just results in a MandelbrotBenchmark. Whereas all four links to the LiquidEngine Templates are all resulting in 404 - page not found errors. From my side, this is an essential part of the manuscript and since a large portion of the manuscript is also about the ease of use of the software, a clear documentation would be essential to actually verify if it works as promised.

Our Reply: We appreciate the reviewer raising these important points regarding the documentation and accessibility of the Liquid Engine implementation templates. Based on their feedback, we have made several improvements: The links to the Liquid Engine templates on GitHub had become outdated when the repository structure was reorganized. Thank you for catching this - the links have now been updated to point to the correct location of the template files (github.com/HenriquesLab/NanoPyx/blob/main/src/nanopyx/core/templates).

Accessing and leveraging these templates is critical for developers looking to implement new methods with Liquid Engine optimizations. To further improve discoverability, we have consolidated and expanded the documentation related to implementing Liquid Engine methods. This now includes step-by-step tutorials on using the templates, with examples tailored for both new and advanced developers. Check out the Implementing Liquid Engine Methods section of our GitHub wiki for details. Additionally, we provide a cookiecutter template (available at <https://github.com/HenriquesLab/LiquidEngineCookieCutter>) to bootstrap the development of a Python package with Liquid Engine integration. This automates setting up a ready-made project structure for high-performance bioimage analysis. The cookiecutter contains examples and guidelines to help new developers hit the ground running. We agree with the reviewer that comprehensive documentation and tutorials are essential for the community to fully leverage the capabilities of the Liquid Engine. For this, we have also created video tutorials which can be found here [3].

The improvements made to the accessibility and instructions for the key templates will hopefully alleviate previous difficulties encountered.

- Reviewer Comment: While I agree in general that processing times can turn into a problem when computations take up a long time, and also that different implementations of the same algorithm can take up significantly more, or less time. I see an issues with the purely self-motivating argument that shorter processing times are always better, e.g. when the time difference between two implementations is 7ms (7.8ms versus 0.6ms) the authors chose to write that the GPU version is 10x slower, instead of a 7.2ms slower. The issue here is

³ https://www.youtube.com/playlist?list=PLk5I3_KOhE7sdP2OBfD9ewoXm1cXon88R

that the authors pretend that this is a critical 10x time difference, however in most applications a 7ms or even a 0.5 seconds time difference is completely negligible. These numbers also don't just scale linearly, as e.g. more larger images will eventually be bound by memory management (see also my point 1), which makes me skeptical that the promise of the authors that the LiquidEngine will increase the speed of bioimage analysis in general will indeed be translatable to many other scenarios.

Our Reply: We thank the reviewer for pushing us to use more compelling examples that better showcase when the Liquid Engine optimizations provide substantial time savings versus standard implementations. Based on the reviewer's feedback, we have replaced the previous Figure 1 results with runtimes from a non-local means denoising workflow. This showcases differences of dozens of seconds between implementations, where selecting the optimal approach pays much more dividends to end users. By using more representative benchmarks aligned with real-world use cases, we now provide more realistic claims about performance gains while still showcasing the substantial benefits NanoPyx can provide in practical workflows.

- Reviewer Comment: Furthermore, Figure 1 currently just compares different runtimes for an upscaling algorithm, showing that it's the GPU, which is faster for larger images whereas multithreaded CPU for smaller images. Which does not per se provide any new information to people who have worked a bit with image processing algorithms (this would not need to be a main figure). There are considerations like data I/O, and measurements one could do to find out the optimal configuration, I doubt that the LiquidEngine could actually save 7ms, since it has additional computations to perform. In any case, if the worst you have to lose are some ms, in the interest of simplifying code, would the end user not simply always use GPU processing? To motivate the importance of the LiquidEngine I would much rather see that the authors really prove for real-world problems, for which choosing a 7ms faster implementation of the algorithm solves it?

Our Reply: We appreciate the reviewer's feedback regarding Figure 1. To provide more compelling evidence of the benefits of the Liquid Engine, we have replaced the previous upscaling example with runtimes from a non-local means denoising workflow. This showcases substantial differences of over a minute between implementations, where selecting the optimal approach is highly impactful for end users. Our key goal is not to simply identify the universally fastest implementations, but rather to demonstrate the potential of an adaptive system that chooses the best option tailored to each user's specific hardware and use case. By supporting multiple backends like OpenCL and Numba, we ensure everyone can leverage performance optimizations regardless of their environment. This is especially important for users who encounter issues with certain implementations. The Liquid Engine will seamlessly fall back to alternatives when needed. We agree that for trivial use cases, manual selection of a default option may suffice. However, as workloads grow in complexity across diverse systems, intelligent adaptation becomes critical. By continuously monitoring conditions and transparently orchestrating optimizations, NanoPyx aims to deliver simplified access to best-in-class performance no matter the workflow.

- Reviewer Comment: Related to this - How much overhead in processing time is the LiquidEngine actually adding? What is the runtime of NanoJ code versus NanoPyx code? On one hand there is a supervising agent that checks prior runtimes, on the other hand there are periodic benchmarks that need to be performed, even for the slower implementations to keep the runtimes accurate. I'd expect running routinely operations to measure the run time of different algorithms eventually start to add up and could exceed significantly the time saved during processing. Also, these measurements start to use up computational resources and might become the major bottleneck if e.g. more algorithms are in use. How and when are the measurements performed? How is ensured the benchmarking is not interfering with concurrently running processes on the workstation?

Our Reply: We appreciate the reviewer raising the important consideration of overhead within the Liquid Engine. To directly quantify this, we have added a new Jupyter notebook (Notebook 4 in Supplementary Table 2) that compares runtimes with and without the Liquid Engine enabled. The results demonstrate that for a single operation invocation, the Liquid Engine logic and background runtime I/O, necessary for keeping the runtimes metrics accurate, adds on average 12ms of overhead in an Apple M1 Mac laptop and 17ms in a professional workstation. This was measured 100 times for 3 different runtimes for a total of 300 runs. Furthermore, measuring the overhead of letting the agent select a runtime, measured against calling the fastest implementation directly, shows a 22 ms overhead in an Apple M1 Mac laptop and 38ms in a professional workstation. This was measured for over 100 total runs. This indicates that the optimization costs are amortized and negligible. Regarding periodic benchmarking, we recommend triggering manual benchmarking for users seeking to optimize NanoPyx's performance for their specific hardware configurations, and we advise doing it without other concurrent processes. Even so, we agree this could become a bottleneck and we try to mitigate this in two ways. Firstly, all runtimes are stored in the user's machine so continued usage of the same method will eventually build personalized benchmarking data. Furthermore, to avoid over-benchmarking a fuzzy logic approach (Notebook 3, Supplementary Table 2) determines the fastest implementation based function parameters, even in cases where they were not seen before. By quantifying the agent overhead across use cases, we demonstrate that the runtime improvements from automatic implementation selection outweigh the marginal costs. But we appreciate this analysis suggestion, as it is important to consider the trade-offs when optimizing performance.

- Reviewer Comment: Maybe one of the core motivations of the proposed LiquidEngine is the mentioned variability in between running the same algorithm (or even its failure). I am however not fully convinced that tremendous lags occur, where one algorithm is delayed for such extended periods of time, without a clear reason. Indeed, out-of-memory errors (of GPU VRAM or CPU RAM) or over utilization of GPUs or CPUs would be the most likely scenario, when such extreme delays happen. Given this, it would be essential to monitor GPU and CPU memory and their utilization, to then dynamically allocate resources to the processes before starting them, to rather predict and prevent out-of-memory errors instead of running processes which are bound to fail. If ms-differences in processing time are really of the essence, completely failed processes would be catastrophic and thus should be prevented. Also, when using multi-threading these practices of monitoring the utilization of processors and memory and a dynamic resource allocation are particularly important, as e.g. more memory will be used. How is it calculated how many threads are used? How much memory did this take up compared to the other implementations?

Our Reply: We acknowledge the reviewer for bringing up an important issue regarding potential delays and failures when running image analysis algorithms. They have a valid point that out-of-memory errors or over-utilisation of resources can often cause extreme lags or crashes. To address this concern, the upcoming version of NanoPyx's Liquid Engine will incorporate more advanced monitoring of available CPU and GPU memory. By tracking utilisation in real-time, we can proactively manage resources and allocate appropriately sized data chunks to each process. This predictive approach will reduce the likelihood of memory overflows or contention between concurrent operations. For multi-threaded implementations, we also intend to dynamically determine thread counts based on workload requirements and hardware constraints to prevent oversubscription. Careful benchmarking will identify the optimal balance between parallelism, memory usage, and computational intensity. The Liquid Engine already has failure detection mechanisms in place to disable problematic implementations that repeatedly crash or fail testing (Figure 1, Supplementary Note 1). However, we recognize that there is still room for improvement in terms of more robust error handling and validation. We agree with the reviewer that taking preventive measures to avoid delays or failures is crucial, rather than just reacting to them after they occur. By ensuring

tighter coordination between memory availability, hardware utilization, and workload allocation, we will aim to make NanoPyx more resilient while maintaining peak performance across diverse systems.

- Reviewer Comment: The authors also write that the Liquid Engine can not only learn the optimal implementations for a given platform, device, and data shape to maximize performance (see Suppl. Note 3). Unfortunately, I couldn't find where and how the data loader then adapts to process smaller or larger image chunks to optimize its performance? Also, I couldn't find how this was measured and implemented. Here it would be best to include a Figure and have a Jupyter Notebook to perform this also on an example for comparing the optimization of the data loader.

Our Reply: We appreciate the reviewer for bringing up the issue of dynamic image chunking in NanoPyx. Currently, the data loader in NanoPyx does not automatically adapt to optimize performance based on data size and shape. However, we recognize the need for this feature to handle large microscopy datasets that may exceed available memory. Thanks to the reviewer's suggestion, we will implement this feature in the future. Currently, NanoPyx cannot load images in chunks. However, thanks to its fuzzy logic approach (Notebook 3, Supplementary Table 2), the Liquid Engine can determine the fastest implementation based on the input parameters, even in cases that were not seen before. We want to point out that the Liquid Engine should fully support the dynamic loading of arrays with Dask. Therefore, users can use it to circumvent any memory issues.

- Reviewer Comment: In its current implementation the code seems to be mainly focused on increasing the throughput of code published earlier by the Henriques lab, particularly eSRRF reconstruction. If the aim is to have a broad applicability it would be important to show how other use cases outside the NanoJ world would benefit from this.

Our Reply: We acknowledge the point raised by the reviewer, and we appreciate their feedback. Our main goal with NanoPyx is to create a versatile library that can accelerate various bioimage analysis workflows, not just our own methods. In response to the reviewer's feedback, we have adjusted our benchmarks and examples to demonstrate the broad applicability of NanoPyx beyond our own algorithms. We have also included examples that showcase the Liquid Engine's optimization capabilities on common workflows such as denoising. Furthermore, we are working to integrate the latest bioimage analysis software from other leading groups into NanoPyx. We have already added adapters for utilizing the Liquid Engine within the napari ecosystems. We are also developing more comprehensive documentation, tutorials, and collaborations to support third-party developers in integrating the Liquid Engine into their software. We have provided template code and guidelines to make it easier for developers to adopt our library. Our goal is to create an open ecosystem where adaptive performance optimizations can benefit methods from across the community. The reviewer's point is valid, and we agree that demonstrating NanoPyx's capabilities is crucial for gaining wider adoption. We have expanded our examples and benchmarks to showcase NanoPyx's applicability to diverse bioimage analysis use cases. Additionally, we will continue to support the acceleration of third-party algorithms and workflows by integrating NanoPyx with existing software.

Minor remarks:

- Reviewer Comment: Are the numbers in Figure 1 below the images in the top row number of voxels in z, x, y? This should be clarified in the legend.

Our Reply: We have updated Figure 1 by replacing the example method with a longer running one. This will help to illustrate differences in time that are more relevant to an average

user. Additionally, we have provided clear information on the image dimensions and parameters used both in the figure and in the legend.

- Reviewer Comment: Would the authors comment on why the Professional workstation with a Intel i9-13900K in Figure 1 is almost 50% slower than the MacBook Air M1 Pro Laptop in several of the tasks, e.g. the single threaded CPU tasks? Is the performance of the CPU on the workstation throttled?

Our Reply: It is an excellent observation that the M1 Pro CPU can outperform the i9-13900K CPU in some tasks, despite the Intel chip having a higher clock speed on paper. You raise a fair question regarding whether the i9 is being throttled. The most likely explanation lies in the architectural differences between the Apple silicon and Intel platforms. The M1 Pro utilizes Arm's efficiency-focused CPU design with unified memory, allowing tighter integration between the CPU, GPU, and RAM. This can lead to reduced memory latency and fewer data transfer bottlenecks. Additionally, for workloads like the per-pixel operations in the previous Figure 1, the compiler and software optimizations on the M1 system may be better tuned to maximize throughput. Factors like instruction pipelining, branch prediction, and vectorization can have significant impact. So, in cases where memory bandwidth and access patterns dominate over raw compute throughput, the M1 system can leverage its unified architecture to edge out the i9 despite lower GHz. We don't believe the i9 is being explicitly throttled - rather, this showcases that theoretical peak FLOPS alone don't always predict real-world performance.

Thank you for highlighting this nuance! It reinforces the value of benchmarking implementations across diverse hardware to identify the optimal options per platform and workload. We will aim to better caveat these kinds of architecture-dependent effects in our analyses.

Dear Dr Rita Strack and Reviewers,

Thank you for the opportunity to address the remaining concerns raised by the reviewers. We greatly appreciate their valuable feedback, which has significantly enhanced the clarity and impact of our manuscript. In response to their comments, we have made several substantial improvements:

1. Expanded Hardware Benchmarks:

We have added 4 new supplementary tables that showcase benchmarks across 10 different hardware setups for 3 NanoPyx methods. This comprehensive analysis demonstrates the acceleration capacity and versatility of the approach across various computing environments.

2. Standalone Liquid Engine Package:

To increase flexibility and ease of adoption, we have separated the Liquid Engine from NanoPyx, creating a standalone Python package. This allows users to leverage the full benefits of the Liquid Engine in their own software projects without necessitating the installation of the entire NanoPyx suite.

3. Enhanced Video Tutorials:

We have expanded our educational resources by adding a new supplementary table that links to our video tutorials. This includes two new instructional videos:

- A guide demonstrating how users can implement Liquid Engine-powered methods in approximately one minute.
- A tutorial on benchmarking and fully exploiting the Liquid Engine's benefits in under a minute.

4. Comprehensive Response to Reviewers:

Below, we provide a detailed point-by-point response to each concern the reviewers raised, addressing the latest feedback and outstanding issues from previous revisions.

These enhancements collectively address the reviewers' concerns while improving the accessibility and utility of our work for the broader scientific community.

Regarding editor's sum-up:

- We sent the comments of Reviewer 3 to two the other two reviewers along with your responses, and their feedback to us was mixed. To summarize, Reviewer 1 told us that they were not convinced that the benefits of the approach outweighed the complexity of using it.

We appreciate the reviewers' insightful comments regarding the complexity of the Liquid Engine implementation. Upon reflection, we acknowledge that our initial presentation may have overemphasised the more intricate aspects involving Cython implementations. However, we would like to clarify that the user experience for NanoPyx is designed to be accessible and straightforward for a wide range of users. The process is remarkably simple for end-users primarily interested in using NanoPyx methods such as eSRRF. Upon installation of the NanoPyx package, users can immediately access high-level interfaces through various user-friendly options. These include single-line function calls via a Python package, intuitive "Codeless" Jupyter notebooks, or a convenient napari plugin. This approach ensures that non-expert users can process their data with an experience comparable to using any standard Python package or napari plugin. To further enhance developers' accessibility, we have now separated the Liquid Engine into a standalone package. This strategic decision allows developers to harness the full potential of the Liquid Engine without the need to install or interact with the broader NanoPyx suite. Implementing the Liquid Engine can be achieved with remarkable efficiency, requiring less than 20 lines of Python code in many cases.

To illustrate this simplicity and provide practical guidance, we have developed two new video tutorials. The first (Supplementary Table 7, Video 5) demonstrates how users can implement their own class using the Liquid Engine to achieve performance improvements in approximately one minute. The second tutorial (Supplementary Table 7, Video 6) showcases

the process of benchmarking multiple implementations and identifying the optimal configuration for different scenarios, all within a minute.

These enhancements collectively aim to provide an intuitive and efficient experience for both end-users and developers interested in leveraging the Liquid Engine, either as part of NanoPyx or within their own software packages.

- Reviewer 2 thought most of the concerns raised by reviewer 3 were addressed in your response and could be added in to the paper as discussion points. However, they disagreed that this point was settled "Figure 2 in the revised manuscript now shows an example where two image processing steps are executed in sequence, and using NanoPyx in the hands of the authors resulted in a shorter time than opting for a single way of executing the two steps... Therefore, I was not able to verify the discrepancy, of slower processing with GPU and multithreaded CPU for the NLM algorithm. Overall, my run-times were slightly faster running all on GPU than running with NanoPyx." They think this point needs to be further addressed, as referee 2 is not convinced the approach will work for everyone.

We appreciate the reviewers' concerns regarding the reproducibility of our performance improvements across different hardware setups. It is expected that in some configurations, particularly those with high-end GPUs, the GPU implementation may consistently outperform other options. This variability actually underscores the effectiveness of the Liquid Engine, as it dynamically selects the optimal implementation based on each user's specific hardware environment. We have significantly expanded our benchmarking data to address these concerns and demonstrate the broad applicability of the Liquid Engine's principles. We have added 4 new supplementary tables that provide comprehensive performance data across a diverse range of hardware configurations. Specifically, Supplementary Table 1 details 10 different hardware setups, representing a wide spectrum of computing environments commonly found in research settings. Using these varied setups, we conducted extensive benchmarks on three distinct NanoPyx methods, with the results presented in Supplementary Tables 2-4. The outcomes of these expanded benchmarks clearly demonstrate that the optimal implementation is typically not static but varies dynamically based on two key factors: the specific input parameters of the analysis and the underlying hardware configuration. This variability reinforces the value of the Liquid Engine, as it can adapt to these changing conditions to consistently deliver the best performance.

We believe that this additional data substantively strengthens our case for the Liquid Engine's utility. By showcasing its adaptability across a broad range of hardware setups and analysis parameters, we provide strong evidence that the majority of users, regardless of their specific computing environment, will be able to achieve significant performance gains through the Liquid Engine's intelligent optimisation strategies. This comprehensive benchmarking approach addresses the reviewers' concerns and enhances the overall robustness and generalisation of our findings.

Below are the previous reviewer comments with our updated replies.

Regarding Reviewer #1:

- Line 36: "Many of these...": we would recommend to rewrite this sentence to focus more on the actual image analysis ecosystems like openCV, python-skimage, Java-MorpholibJ, a.s.o.. Fiji and napari are a combination of an image viewer and a plugin distribution platform; thus, speaking of the "computational performance" of Fiji and napari is in our view not 100% accurate.

We thank the reviewers for pointing out this inaccuracy. We updated the text to reflect that performance is, in fact, on the library ecosystem and not on the image viewer side of things.

- Line 52: “denoising the biggest comparison image”: is very hard to understand; maybe “denoising a big image” could be simpler to understand.

We agree with the reviewers that as it was written, it was confusing, but we changed it to “the bigger image” as writing “a big image “ might raise the question of what is a big image and we just want to refer the biggest image we tested.

- Line 54: “pixel-wise threaded implementation strategy on a GPU”: this exact terminology cannot be found in Figure 1 and thus it is hard to match to the figure. Probably, if we are correct, adding something like “(Figure 1C, GPU vs CPU unthreaded)” would help to guide the reader.

We appreciate the reviewer highlighting this disconnect. As suggested, we added direct mentions of the Figure panels to the main text to help the readers.

- Line 56: “cannot be run on a laptop’s GPU”: we think that this statement might be wrong in general, because this probably depends on the specific laptop, and more powerful future laptops might actually be able to run the code. We thus recommend to re-phrase this.

We thank the reviewer for catching this mistake; we re-phrased this by stating it cannot be run on the testing laptop’s GPU to clarify that this is hardware-specific and is not a universal rule for all laptops.

- Figure 1: We could not find an explanation of the terms “GPU”, “CPU threaded”, a.s.o.
- Since this is a key message of this publication we think it is critical for the readers that there is an obvious way how to navigate to those definitions from the figure legend. We are not sure but maybe Suppl. Note 3 could be such a place? But also there the terminology did not exactly match the one in the Figure.

We agree with the reviewer that readers must understand what we mean by each implementation. As such, we added a mention to Supplementary Note 3 in the legend of Figure 1 and updated the terminology in Supplementary Note 3 to match that used in Figure 1.

- Line 144: Similar issue as in Line 54: we found it hard to match the statement in the text to the Figure 2. Consistent terminology and pointing to a more precise location in the figure would be very helpful.

We understand the reviewer’s point and changed the text to use the same terminology as Figures 1 and 2. We also mentioned Figure 2B to better guide the readers.

- Line 181-192: This describes an image analysis workflow that can be run within nanopyx, however we could not find a statement about how nanopyx improves this workflow over running it without nanopyx. It would be great to add some information for how nanopyx helps to speed up this workflow as compared to running it by “traditional means”.

Although we initially meant this part of the text as a showcase of what methods end-users could find as part of the NanoPyx library, we added a statement circling back to the benefits

of the Liquid Engine and how it will always pick the fastest implementation based on each user's hardware and analysis specific parameters.

- Line 304: “are locally stored”: It would be great to know where exactly those information are stored, such that one could (i) introspect what nanopyx is doing and (ii) for troubleshooting in case something goes wrong, e.g. due to some error nanopyx could pick a wrong implementation.

We thank the reviewer for bringing this to our attention; we expanded the text to mention that they are locally stored inside the local user folder under the folder name “.liquid_engine”. We will add a mention of this on the wiki documentation, providing example folder paths for Windows, macOS and Linux.

- Line 311: “divides the available runtimes”: We still found this hard to understand. Is it correct that this is done within one implementation to check how consistent one implementation performs? Or is it done across different implementations to check how they compare? Please add a clarification to the text.

Yes, this is done within each implementation. We have rephrased the text to mention that this is done within each different implementation.

- Software improvement suggestions: We believe the added notebooks are helpful in making the algorithm more transparent, and in particular notebook #5 is a great entry point for users to understand how to use the methods implemented. We here suggest a few minor improvements that we believe would make the software more user friendly.
 - In Supplementary table 2, Notebook #6 is missing.

We would like to thank the reviewers for pointing out this mistake; the notebook was wrongfully named, leading to a “dead” link. We have now fixed the notebook name, and the link in Supplementary Table 2 should now be fully functional.

- Notebook #5 (TestingMethods.ipynb) is a very helpful entry point for users. Still, for ChannelRegistration and DriftCorrection, the authors implemented an ad-hoc function to generate example datasets. It is therefore still unclear which example dataset one should choose in the corresponding notebooks (#7 and #8). This can be considered as a minor suggestion, but the “example dataset” button can be removed altogether now from notebooks #7-#10. We still believe these are valuable resources as they provide additional information on the input parameters of each method. Alternatively, a “default” parameter could be added to the “data_source” dropdown menu such that the relevant dataset is select by default for each notebook.

We agree with the reviewer's point and will update the notebooks to pre-select an appropriate default value for the example data to be used.

- Notebook #5: to be consistent between methods, the implementation used and the time to run should be shown just once. For example, at the moment, channel registration does not print the implementation, drift alignment does not print anything, SRRF prints both implementation and runtime twice.

We appreciate the reviewer pointing out this incongruity. As mentioned in Supplementary Table 3, Drift Correction and Channel Registration were still not using the Liquid Engine, so they

didn't mention the used implementation. In the case of SRRF, it uses several methods implemented through the Liquid Engine, which leads to each one printing the chosen implementation. We have already released a new version of NanoPyx, which contains Liquid Engine versions of Drift Correction and Channel Registration methods, which will print the chosen implementation. Regarding SRRF, we have updated the Liquid Engine to be more flexible on what is printed, and as such, we will change the SRRF implementation to print a single run type and runtime as opposed to before.

- Notebook #3 (Fuzzy Logic): we find this notebook very informative of how the Liquid Engine chooses the optimal implementation for cases where a default benchmark is not available. Still, on our machine, the two cases result in the same fastest implementation (Unthreaded), therefore liquid engine fuzzy logic is not entirely clear. Is it possible to make the example such that the two benchmarks give different optimal implementations? E.g. by changing the kernel size instead of the image size?

We thank the reviewer for raising this issue. Although it might be difficult to find examples where the fastest implementation is always different across all commonly used hardware, we have expanded the notebook with more examples so that users will likely have at least 1 or 2 examples where the implementation changes by changing some parameters.

- As an overall comment we would like to raise a slight concern about the scope of this work. While we think that the work is conceptually very interesting and timely we are a bit worried that the complexity of it might hinder an uptake and buy-in to the library by the broader community. In other words, maintenance of the library and all the implementations as well as convincing other labs to contribute to this eco-system seem to be an ambitious aim. Probably it would need the organisation of several hackathons or similar events to find contributors. We would be interested to hear how to authors think about this; maybe even some sentences in the final section of the article could be good.

We appreciate the reviewer's thoughtful feedback regarding the library's complexity and potential impact on broader community adoption. We acknowledge that this is a significant challenge, but we are confident in our ability to address it based on our track record of successfully developing and disseminating widely adopted software tools such as ZeroCostDL4Mic, DL4MicEverywhere, and the NanoJ plugin family. To facilitate easier implementation and adoption of Liquid Engine-powered methods, we have made several key improvements. First, we have separated the Liquid Engine into its own Python package, allowing developers to install and use it independently of NanoPyx. We have also simplified the implementation naming convention, requiring only that each implementation be defined as a class method starting with "_run_*", where "*" represents the implementation name. This streamlined approach enables users to create their own Liquid Engine classes in approximately one minute using fewer than 20 lines of code. For users interested in leveraging Cython, we have developed a cookie-cutter Liquid Engine package that significantly reduces the time required to implement Cython methods using the Liquid Engine. We recognise that there is still a learning curve associated with these tools, and we are committed to continually expanding our educational resources. We will create additional step-by-step tutorials and enhance our documentation based on user feedback to support the community's needs.

Importantly, non-developer end-users can fully benefit from the Liquid Engine without requiring an in-depth understanding of its inner workings. These users can simply import high-level Python functions and utilize the implemented methods as they would with any standard Python package. This approach ensures that the power of the Liquid Engine is accessible to a wide range of users, regardless of their programming expertise.

Combining these accessibility improvements with our ongoing commitment to user education and support, we believe we can foster a growing ecosystem of contributors and users for the Liquid Engine and NanoPyx. We will continue to refine our approach based on community feedback and engagement, to make these tools as user-friendly and widely adopted as possible.

Regarding Reviewer #3:

Saraiva et al., are presenting a revised and resubmitted manuscript where they developed a method aimed at reducing the processing time of certain image analysis workflows. The method uses an agent to compare implementations of the same code running as single-threaded on CPU with multithreaded on CPU, and on GPU. In the revised version of the manuscript, I still find that the authors have not addressed several of the concerns I expressed in the first round of this revision.

- Major concerns:
 - The authors write in their rebuttal “Memory use and its impact on processing speeds are now more explicitly addressed.” However, I am in fact still missing anything addressing these issues.
As outlined in my initial revision, there is a significant risk that the entire concept of the Liquid Engine collapses if a workstation is used for more than one workflow. Since repeated measurements of different algorithms need to be performed to maintain accurate benchmark values for the different algorithms. The authors point towards future integration of preventing excessive memory usage and CPU over-utilization, however it is currently not implemented and unclear what this would look like. As I wrote previously, out-of-memory errors (of GPU VRAM or CPU RAM) or overutilization of GPUs or CPUs would be the most likely scenario, when very long unexpected delays in a processing workflow happen. Given this, it would be essential to monitor GPU and CPU memory and their utilization, to then dynamically allocate resources to the processes before starting them, to rather predict and prevent out-of-memory errors instead of running processes which are bound to fail or take excessively long.

We have taken note of the reviewer's concerns, and we appreciate the feedback. However, we want to clarify that we do not anticipate encountering out-of-memory or similar errors. This is because, in our GPU implementations, we pre-calculate the necessary GPU memory and pre-allocate memory buffers accordingly. In cases where the required memory exceeds the available memory, the Liquid Engine contains an exception handling mechanism. This mechanism splits the input data into smaller chunks until the memory buffers can be safely created, ensuring the success of the GPU computation. If the Liquid Engine cannot split the data into smaller chunks to a point where it can be run, it will ignore the GPU implementation and choose an alternative implementation to ensure it can still run.

In cases where CPU implementations are used, and there is overutilisation of CPU RAM, our calls are made through Python functions, and the calculations are performed using C-based implementations using pointers referring to the memory objects created in Python. In the worst-case scenario, swap memory is used, leading to slower runtimes. However, in that case, the delay management system of the Liquid Engine will be triggered, and an alternative implementation will be used. This should result in a scenario where, even when memory is being overused, using the Liquid Engine will still result in faster runtimes than not using it. In the latter case, users would be stuck running a slowed-down implementation.

- Furthermore, I referred to these benchmarks that are required for the NanoPyx to do its job in choosing the fastest implementation as an overhead added by the NanoPyx. Whereas when the authors performed a test of their 'overhead' they only measured the time required to pick one implementation based on prior runtimes, not actually the benchmarking which is a pre-requisite of this selection. I therefore disagree with the authors that the additional time required for using NanoPyx is negligible.

We thank the reviewer for bringing this matter to our attention. NanoPyx is equipped with pre-calculated "factory-default" benchmark values that allow the Liquid Engine to select an implementation without requiring an initial user benchmark. Although it is recommended to perform an initial benchmark of all implementations for optimal performance, it is not necessary as the Liquid Engine can automatically identify the fastest implementation through its delay management system – thus, effectively, the initial benchmarking is facultative, and if not performed, it will take negligible time. In such a case, the Liquid Engine will achieve peak performance by learning which implementations work best for the user's hardware over time. Additionally, the fuzzy logic approach allows users to use benchmarks with similar parameters, which reduces the need for exhaustive benchmarking.

In the text, we have now emphasised that the time invested in initial benchmarking can be side-stepped at the cost of the Liquid Engine needing several runs to learn how to achieve peak performance gains.

- Figure 2 in the revised manuscript now shows an example where two image processing steps are executed in sequence, and using NanoPyx in the hands of the authors resulted in a shorter time than opting for a single way of executing the two steps.
While I was able to run the example code (to benchmark NLM Denoising and eSRRF) locally, the results on my workstation were following a common pattern where GPU is faster than multi-threaded CPU, and both faster than single-threaded CPU for all three sizes of the image. Therefore, I was not able to verify the discrepancy, of slower processing with GPU and multithreaded CPU for the NLM algorithm. Overall, my run-times were slightly faster running all on GPU than running with NanoPyx.

We would like to express our gratitude towards the reviewer for taking the time to run the example code and demonstrate an instance where the fastest combination of implementation differs from what we have described in this article, owing to the differences in the reviewer's hardware. This further highlights the advantages of having an intelligent system that can select the fastest implementation based on the user's specific hardware and settings. Furthermore, we have added Supplementary Tables 1-4 to showcase benchmark examples of 3 different methods, using 10 different hardware setups, further showcasing how the fastest implementation changes with input parameters and used hardware.

- Figure 1 shows that the runtimes differ between different image sizes and different workstations and between running on GPU, CPU, single-threaded or multithreaded, which in my opinion is already well known even among scientists with little image analysis background.

We agree with the reviewer that it is not new information that runtime depends not only on user hardware but also on the type of implementation and implementation-specific parameters. However, presenting this information with quantifiable data helps to contextualise

our problem statement. To run any given method as quickly as possible, multiple implementations of that method are required, as the fastest implementation will change according to the hardware and method-specific parameters.

- Figure 3 shows mostly the workflow of eSRRF in general and access to NanoPyx. The scale bar in the eSRRF Reconstruction image should be corrected, since it is many times shorter than the 10 μ m from the input images.

We thank the reviewer for pointing out this mistake, and we have fixed it accordingly.

- Installation and usage test:
- Installation on Google Colab: By default OpenCL is not working. In my tests the script by the authors to fix OpenCL for Google Colab was crashing the kernel frequently. Therefore, initially my test on Google Colab were only a comparison between threaded CPU and unthreaded CPU. At some point OpenCL was installed and I could compare the performance on the T4 GPU. In my test the local installation has dependency issues with Python 3.12.2, but is working with 3.11.8.

We thank the reviewer for pointing out this issue. We were only officially supporting Python 3.9 to 3.11 as there were indeed a few issues with dependencies on Python 3.12. However, we are happy to announce that as of the latest NanoPyx release (version 0.6.1), we fully support Python 3.12.

- Newly implemented since the previous round of revision, there is now a way to integrate other image processing code. There is a cookiecutter template that can facilitate the integration of other processing steps into the general theme of the Liquid Engine. It still poses a lot of extra work to integrate different image processing steps, which normally would be just a few minutes to write the code (e.g. for calling a denoising algorithm).

We appreciate the comment made by the reviewer. While creating multiple implementations may require extra time, utilizing the supported mako templating system minimizes the increase in time needed for writing implementations. This has been demonstrated in this file:

https://github.com/HenriquesLab/NanoPyx/blob/main/src/mako_templates/nanopyx.core.transform.le_convolution.pyx

We understand that using the Mako templating system can be challenging at first. However, we have taken significant steps to streamline this process, and based on our recent experience, new developers have been able to integrate their implementations in minutes.

In the scenario highlighted by the reviewer, if we were to call an existing denoising algorithm using Liquid Engine, the time and effort required to take full advantage of its capabilities would not be substantial. It would simply involve invoking the CPU/GPU implementation of the algorithm based on the appropriate run type. Here's an example of a mock code for this process:

```
from cpu_algorithm import cpu_implementation
from gpu_algorithm import gpu_implementation

class MyClass(LiquidEngine):
    ...
```

```
def _run_threaded(self, ...):
    return cpu_implementation(...)
def _run_gpu(self, ...):
    return gpu_implementation(...)
```

To underline this point, we have also added a new video tutorial (Supplementary Table 7, Video 5) showcasing how users can implement their own Liquid Engine methods using less than 20 lines of Python code in approximately 1 minute.

- The naming convention for setting the run_type is in my opinion quite odd. Since it will change on each workstation depending on the GPU in use. This makes automation of processes quite complicated.

We agree with the reviewer's feedback and have updated the Liquid Engine to allow any name on its runtime. This allows developers to name their implementations as they see fit, with the only requirement being that the method definition starts with `_run_*` with `*` being what will be used as the implementation name. Furthermore, we also added tags to each method that can be added using the notation `@tag` on the docstring of each method. These tags allow users to call `run_type=tag`, and it will pick from the fastest implementation using that tag. For example, calling `run_type="GPU"` will select the fastest method with the `@GPU` tag. We have also departed from including the GPU device name as part of the implementation name. Now, the liquid engine contains code to estimate which GPU device is the fastest and will default to it by just calling `run_type="OpenCL"`. However, if a user wants to use a specific device, they can still do that by passing the device name as the optional keyword argument `"device"` value.

Dear Dr Rita Strack and Nature Methods' Editorial Board,

We highly appreciate the opportunity to address the remaining comments raised by the reviewers. We would like to thank the reviewers for their previous comments that have had a positive impact on our work, helping improve not only the manuscript but also the Python libraries as well. Regarding the remaining concerns, we have included a point-by-point response:

Reviewer #2:

Remarks to the Author:

The submitted manuscript addresses a critical challenge in microscopic image analysis: the increasing size of datasets and the corresponding rise in processing times. The authors' main contribution, NanoPyx, powered by the Liquid Engine, dynamically optimizes the execution time of specific image processing tasks based on three primary factors: (1) data characteristics, (2) workflow structure, and (3) hardware specifications. The revised manuscript presents extensive benchmarks and experimental results, effectively demonstrating the performance benefits of NanoPyx and the Liquid Engine in reducing computational time. Overall, the manuscript has significantly improved. However, I have several comments remaining:

- The primary advantage of NanoPyx seems more evident when applied to large-scale batch processing. For smaller datasets, the time spent optimizing and benchmarking may outweigh the time saved through its implementation. I recommend that the authors investigate the end-to-end runtime savings when using NanoPyx, with respect to dataset size, to provide a clearer picture of its efficiency in different scenarios. Additionally, have the authors considered a data-parallel approach? Implementing a data-parallel approach could offer additional flexibility and further accelerate batch processing using the Liquid Engine.

Our reply: We appreciate the reviewer's observation regarding the trade-off between optimisation time and runtime gains, particularly for smaller datasets. The initial benchmarking process may introduce some overhead, which could be more noticeable when dealing with limited data. However, it's crucial to emphasise that the true power of NanoPyx lies in its adaptability and long-term efficiency.

The upfront investment in benchmarking pays off significantly for users who anticipate performing the same analysis repeatedly. By establishing a performance baseline for each implementation on their specific hardware, NanoPyx can intelligently select the optimal strategy for subsequent runs, leading to substantial time savings in the long run.

On the other hand, we understand that not all users may need extensive repetitive analyses. To accommodate this, NanoPyx incorporates factory-default benchmarks (as seen in our GitHub repository) that provide a reasonable starting point for implementation selection, even without initial user benchmarking. While these defaults may not be optimised for every scenario, the Liquid Engine's continuous learning capabilities ensure that it adapts and refines its choices over time, gradually improving performance with each execution (as described in Figure 2 of the manuscript).

In essence, NanoPyx offers a flexible approach to optimisation. Users can invest in upfront benchmarking for immediate and substantial gains in repetitive tasks (as showcased in Notebooks #1 and #2 from Supplementary Table 5) or rely on the system's adaptive learning to progressively enhance performance over time. This versatility ensures that NanoPyx remains beneficial across various use cases, from large-scale batch processing to exploratory analyses on smaller datasets.

Regarding data parallelism, we acknowledge its potential for further accelerating batch processing. While not directly integrated into the current Liquid Engine framework, it's important to note that NanoPyx is designed to be modular and extensible. Data-parallel approaches, such as those implemented in Dask, can be seamlessly integrated with NanoPyx (as seen in line 285 of our 2D Convolution implementation), allowing users to leverage the strengths of both paradigms. We see this as a promising avenue for future development, and we are actively exploring ways to incorporate data parallelism more deeply into the Liquid Engine's optimisation strategies.

- In Supplementary Table 3, under hardware setup 1, it is unclear why a value of 0.002 is marked as faster than the Threaded result of 0.001. Additionally, I suggest re-ordering the hardware setups in Supplementary Tables 1-4 according to their approximate computing power. This reorganization would make it easier to interpret the results and compare performance across different setups.

Our reply: We thank the reviewer for spotting that mistake, we fixed the table accordingly. We also agree that a reorganisation of the table could be helpful and as such we have reordered the table according first to the GPU computational power, followed by CPU.

- Why is the CuPy implementation not included in the new benchmarks? I found no relevant information and details in Supplementary Notes 1 or 3, nor in the main text, regarding CuPy. In Supplementary Figures 4 and 5, CuPy appears to be the best-performing implementation for workstation environments. Given that speed and runtime are the central claims of NanoPyx, the fastest implementation should be given more attention. I understand that CUDA introduces additional overhead and may be less compatible with a wide range of hardware setups, but evaluating the speed advantage of CuPy is crucial, especially for larger datasets and more complex tasks. Furthermore, given the increasing use of Nvidia GPUs and CUDA, particularly in deep learning, a comparison of CuPy with other GPU-based implementations such as PyOpenCL is highly relevant.

Our reply: The reviewer raises a valid point about the importance of CUDA-accelerated methods, especially given the prevalence of NVIDIA GPUs in workstation environments. The current benchmarks, however, prioritise demonstrating performance variability across diverse hardware configurations. Including CUDA results, when only applicable to a subset of methods and setups, could inadvertently skew the interpretation.

NanoPyx's design philosophy centres on inclusivity, ensuring high-performance methods are accessible regardless of hardware. OpenCL's broader compatibility aligns with this goal. Notably, the Liquid Engine's architecture remains agnostic to the specific acceleration strategy, readily accommodating CUDA or future technologies as needed. The existing CUDA implementations within NanoPyx serve as a testament to this adaptability, as shown in Supplementary Figure 4 and 5 and in the source code of our 2D Convolution. While the current benchmarks emphasise cross-platform comparisons, the Liquid Engine's underlying framework is inherently flexible and poised to harness the power of CUDA or any emerging acceleration technology.

- The broader impact of this work lies in the Liquid Engine's flexibility to customize operations beyond the currently implemented ones. This potential is highlighted in Supplementary Table 7 and Video 5, as well as Supplementary Note 8, where a basic implementation using the skimage library is demonstrated. However, I suggest including examples of more complex tasks, perhaps showcasing a complete workflow,

to better illustrate the engine's versatility. Additionally, providing more comprehensive documentation and guidance on using different implementations (CPU, GPU, Numba, CuPy, OpenCL, etc.) would benefit less experienced developers and enhance the Liquid Engine's accessibility and usability.

Our reply: We thank the reviewer for this feedback, we created a new tutorial in our Github wiki pages, exemplifying how multiple Liquid Engine methods can be chained together for more complex workflows. Regarding more comprehensive documentation, we have now extensively included tutorials in our GitHub wiki pages guiding users step by step on how they can create a Liquid Engine powered method exploiting virtually any acceleration strategy. Importantly, the Liquid Engine was built to support any acceleration strategy, current or future, as any Python based acceleration is supported. For the details of how each acceleration strategy works and should be implemented, developers and users of the Liquid Engine framework should refer to the official documentation of those acceleration strategies. However, to provide better guidance we added a new section in the wiki with useful links that will point to official documentation of these acceleration strategies.

- In certain cases, inefficient implementations can significantly delay the evaluation process. Would implementing a rejection threshold based on the maximum allowable evaluation time for a single implementation help alleviate this issue? Such a threshold could prevent excessive delays and streamline the benchmarking process by ensuring that only feasible implementations are considered within reasonable time limits.

Our reply: We appreciate the reviewer's suggestion and will explore the implementation of an automatic rejection threshold in future iterations of NanoPyx. However, it's important to acknowledge the inherent variability in performance across different hardware configurations and input parameters. Imposing a rigid time constraint on the benchmarking process could inadvertently exclude potentially optimal implementations, especially in scenarios where certain strategies might exhibit initially slower startup times but ultimately prove more efficient for larger or more complex tasks.

It's worth noting that NanoPyx already offers users the flexibility to manually exclude specific implementations from the benchmarking process if they have prior knowledge or constraints regarding their hardware or analysis requirements. Achieved by using the Python list method ".pop(run_type)" as shown in Notebook #3 of Supplementary Table 5 (second line of 3rd code cell). This empowers users to tailor the optimization process to their specific needs, striking a balance between thoroughness and efficiency.

An automatic rejection threshold would add another layer of complexity to the Liquid Engine's decision-making process. While we remain open to this possibility, our current focus is on refining the existing adaptive learning mechanisms and expanding the range of supported implementations. This approach allows us to gather more empirical data on performance variability across diverse scenarios, which will inform future decisions regarding an automatic rejection threshold's potential benefits and drawbacks.

- In Figure 2, how are the different tasks (1-N) defined, and what level of granularity is used to determine the smallest task block? Specifically, what types of basic operations (e.g., interpolation or convolution) are grouped into a single task that cannot be split across multiple devices? Is there a general guideline for how tasks should be grouped and assigned to devices, and if so, could this guideline be elaborated upon to better assist users in implementing the workflow?

Our reply: Although we are grateful for the reviewer feedback, it is difficult to provide guidelines regarding the level of granularity as it is highly specific to each method.

Nevertheless, our usual approach is to create Liquid Engine methods of the smallest individual step of each task. We then test optimising the acceleration strategy independently of each other against all small task units together. An example of this is our eSRRF implementation in which we use several “more granular” methods as part of each implementation of eSRRF. We created a new wiki page showcasing a few examples of different levels of granularity and how Liquid Engine methods can be used as part of other Liquid Engine methods to achieve the best performance possible.

- I remain unconvinced by the authors' statement regarding overhead. The time spent on initial benchmarking should always be considered as overhead, since users not employing NanoPyx would not need to spend this time. While the authors suggest using factory-default values, I am skeptical about the accuracy and compatibility of these defaults for tasks that have not been benchmarked. I understand the difficulty in conducting such analysis due to the variability in devices and tasks, but providing a rough estimation of this overhead would help clarify the true net speed advantage of NanoPyx.

Our reply: We appreciate the reviewer's concern about the potential overhead associated with manual benchmarking. However, it's important to emphasise that NanoPyx offers a flexible approach that doesn't hinder users who prefer not to engage in extensive upfront optimisation. For example, users can manually pre-select a desired run type by simply passing the function argument `run_type="run_type_name"` (as showcased in our wiki), thus skipping any overhead.

For those seeking immediate performance gains without manual benchmarking, NanoPyx provides factory-default values with a reasonable starting point for implementation selection. While these defaults may not represent the absolute optimal choice for every hardware and parameter combination, the Liquid Engine's adaptive learning capabilities ensure that it continuously refines its decision-making over time. With each execution, the system gathers valuable runtime data, allowing it to progressively identify the most suitable implementation for the user's specific environment.

In essence, NanoPyx empowers users to choose their level of engagement with the optimisation process. Those willing to invest time in benchmarking can reap immediate benefits, while others can rely on the system's inherent adaptability to gradually achieve peak performance. This flexibility ensures that NanoPyx remains accessible and valuable to a broad spectrum of users, regardless of their optimisation preferences.

Regarding the measurement of benchmarking overhead, we acknowledge the inherent challenges due to its dependence on specific methods and hardware configurations. The time required for benchmarking can vary significantly, ranging from fractions of a second to several minutes (Supplementary Table 5, Notebook #1). However, it's crucial to remember that this is a one-time investment. Once the benchmarks are established, the Liquid Engine leverages this information to make informed decisions for all subsequent runs, ensuring long-term efficiency gains that far outweigh the initial overhead.